Microbiology
SPECTRUM

# Genome-Wide Analysis Reveals that PhoP Regulates Pathogenicity in *Riemerella anatipestifer*

Yang Zhang,[a] Ying Wang,[a,d] Yanhao Zhang,[a] Xiangchao Jia,[a] Chenxi Li,[a] Zutao Zhou,[a,b,c] Sishun Hu,[a,b,c] Zili Li[a,b,c]

[a]State Key Laboratory of Agricultural Microbiology, College of Veterinary Medicine, Huazhong Agricultural University, Wuhan, Hubei, China
[b]Key Laboratory of Preventive Veterinary Medicine in Hubei Province, Wuhan, Hubei, China
[c]Key Laboratory of Development of Veterinary Diagnostic Products, Ministry of Agriculture of the People's Republic of China, Wuhan, Hubei, China
[d]College of Veterinary Medicine, Shanxi Agricultural University, Jinzhong, China

Yang Zhang and Ying Wang contributed equally to this work. Author order was determined both alphabetically and in order of increasing seniority.

**ABSTRACT** Duck infectious serositis, also known as *Riemerella anatipestifer* disease, infects domestic ducks, geese, and turkeys and wild birds. However, the regulatory mechanism of its pathogenicity remains unclear. The PhoPR two-component system (TCS) was first reported in Gram-negative bacteria in our previous research and was demonstrated to be involved in virulence and gene expression. Here, DNA affinity purification sequencing (DAP-seq) was applied to further explore the regulation of PhoPR in relation to pathogenicity in *R. anatipestifer*. A conserved motif was identified upstream of 583 candidate target genes which were directly regulated by PhoP. To further confirm the genes which are regulated by PhoR and PhoP, single-gene-deletion strains were constructed. The results of transcriptome analysis using next-generation RNA sequencing showed 136 differentially expressed genes (DEGs) between the Δ*phoP* strain and the wild type (WT) and 183 DEGs between the Δ*phoR* strain and the WT. The candidate target genes of PhoP were further identified by combining transcriptome analysis and DAP-seq, which revealed that the main direct regulons of PhoP are located on the membrane and PhoP is involved in regulating aerotolerance. Using the *in vivo* duck model, the pathogenicity of Δ*phoP* and Δ*phoR* mutants was found to be significantly lower than that of the WT. Together, our findings provide insight into the direct regulation of PhoP and suggest that *phoPR* is essential for the pathogenicity of *R. anatipestifer*. The gene deletion strains are expected to be candidate live vaccine strains of *R. anatipestifer* which can be used as ideal genetic engineering vector strains for the expression of foreign antigens.

**IMPORTANCE** *Riemerella anatipestifer* is a significant pathogen with high mortality in the poultry industry that causes acute septicemia and infectious polyserositis in ducks, chickens, geese, and other avian species. Previously, we characterized the two-component system encoded by *phoPR* and found that *R. anatipestifer* almost completely lost its pathogenicity for ducklings when *phoPR* was deleted. However, the mechanism of PhoPR regulation of virulence in *R. anatipestifer* had not been deeply explored. In this study, we utilized DAP-seq to explore the DNA-binding sites of PhoP as a response regulator in the global genome. Furthermore, *phoP* and *phoR* were deleted separately, and transcriptomics analysis of the corresponding gene deletion strains was performed. We identified a series of directly regulated genes of the PhoPR two-component system. The duckling model showed that both PhoP and PhoR are essential virulence-related factors in *R. anatipestifer*.

**KEYWORDS** DAP-seq, RNA-seq, *Riemerella anatipestifer*, TCS, gene regulation, PhoP, virulence

Address correspondence to Zili Li, lizili@mail.hzau.edu.cn.

The authors declare no conflict of interest.

**D**uck infectious serositis is also known as *Riemerella anatipestifer* disease. It is one of the severe bacterial infectious diseases in the duck industry and is an acute, contact, and septic infectious disease infecting domestic ducks, geese, and turkeys and a variety of poultry and wild birds (1, 2). The pathological symptoms of this disease are characterized by cellulosic pericarditis, perihepatic inflammation, gasbag inflammation, and meningitis. The Gram-negative bacterium *R. anatipestifer* belongs to the family *Flavobacteriaceae* and has at least 21 serotypes (3). The pathogenicity of *R. anatipestifer* is related to metabolic synthesis-related genes, a bicomponent system, a type IX secretion system, and a CRISPR-Cas system. Tian et al. reported that Dps prevents the damage induced by $H_2O_2$ through iron binding and protects *R. anatipestifer* from oxidative stress and host clearance (4). The functional components of the type IX secretion system (T9SS), GldK, and GldM are reportedly related to the movement, protein secretion, and virulence of *R. anatipestifer* (5–7).

The ability of pathogens to sense and respond to environmental changes encountered within the host is generally believed to be essential for sustaining the bacterium's pathogenicity and survival (8). The classic two-component system (TCS) consists of two parts: the histidine kinase (HK)-sensing protein, which is usually membrane bound and acts as an environmental sensor with a signal receiver domain, and the response regulatory protein, which often functions as a transcription regulatory factor with a DNA-binding domain (8). Of these, the phosphorylated histidine kinase-sensing protein phosphorylates the response regulator (RR) in the cytoplasm. The phosphorylated response regulator upregulates or downregulates the expression of the bacterial genes (9).

Discovering the gene regulated by TCSs of bacterial pathogens is essential for understanding the mechanisms of bacterial survival and infection. PhoPR and PhoPQ mainly exist in Gram-positive bacteria and Gram-negative bacteria, respectively, and PhoR and PhoQ showed a high similarity in their C-terminal portion. Although many studies have reported that PhoPR and PhoPQ are required for pathogenicity in their respective species (10–12), they are not physiologically equivalent. PhoPR in Gram-positive bacteria is mainly activated under low-phosphate conditions, regulating gene expression to cope with low-phosphate environments (13), while PhoPR in *Mycobacterium tuberculosis* does not respond to phosphate (14). In Gram-negative bacteria, PhoPQ mainly responds to magnesium limitation and antibacterial peptides (12, 15), and phosphate starvation is sensed by PhoBR (16). A *phoPR* double gene deletion strain of *R. anatipestifer* was constructed in our previous research, and it is the first *phoPR* TCS reported in Gram-negative bacteria (17). Transcriptome sequencing (RNA-seq) using KEGG pathways and the upregulation of PhoPR in phosphate starvation indicated that *RAYM_RS09735-RAYM_RS09740* was the PhoPR two-component system, and the results of differential expression gene analysis showed that the TCS was a global regulatory factor of *R. anatipestifer*. The results of animal experiments showed that the double-gene-deletion strain completely lost its pathogenicity for ducklings (50% lethal dose [$LD_{50}$] > $10^{11}$ CFU), so PhoPR is confirmed to be involved in the virulence of *R. anatipestifer*.

The characteristics of the direct regulatory ability of the RR's DNA binding are generally studied to further explore the regulatory mechanism of TCS in pathogenic bacteria. Generally, HK is phosphorylated after receiving external signals, transferring the phosphorylation group to the effector response regulator, and the phosphorylated RR begins to regulate the target genes. In this study, DNA affinity purification sequencing (DAP-seq) was applied to explore the direct regulation mechanism of PhoP *in vitro* (18–20). This assay allows us to mimic the RR phosphorylation via activation by small-molecule donors like acetyl phosphate (acetyl-P) to find the genes affected directly by the regulators (21, 22).

In this study, *phoP* and *phoR* single-gene deletion strains were constructed based on the construction of the *phoPR* double-deletion strain, and the mechanism of regulating the virulence of *R. anatipestifer* was studied by combining DAP-seq and RNA-seq. Our data provide a foundation for understanding the role of the PhoPR two-

component system in the pathogenic process of *R. anatipestifer*, a new insight into the regulation of PhoP for aerotolerance in *R. anatipestifer*, and a theoretical basis for discovering new drug targets. This direct and global exploration of the regulation of *phoP* provides a model for gene regulation in *R. anatipestifer* and other pathogens. Establishing a model for transcriptional regulation can provide a technical platform for mining new virulence factors, diagnostic markers, and vaccine candidate antigens.

## RESULTS

**Genome-wide identification of PhoP binding sites in *R. anatipestifer* by DAP-seq.** The PhoPR TCS was characterized by next-generation RNA sequencing in our previous study. This indicated that almost 1/3 of genes in *R. anatipestifer* are under the regulation of PhoPR at the transcriptional level (17). While a few genes were chosen previously for testing the PhoP binding of their promoters, the biophysical interactions between PhoP and the promoter regions of the genes controlled remain unclear. DAP-seq was utilized for investigating the PhoP-binding region on the whole genome of *R. anatipestifer*, which could effectively enrich the binding peaks of the phosphorylated PhoP and avoid the disadvantage of lacking specific antibodies and indirect PhoP binding in *vivo*. The previous version of genome assembly consisted of 29 scaffolds (accession number: AENH00000000), so we subsequently resequenced RA-YM via the PacBio platform to generate the completion map. The long read length of third-generation sequencing and the small impact of GC content enabled us to solve the problem of repetitive sequences in RA-YM and avoid the unevenness of sequencing caused by GC content, which is useful for *de novo* genome assembly (23). The new version of the genome is 2,153,508 bp in length, with an average G+C content of 35.02%.

The recombinant His$_6$-PhoP was expressed in *Escherichia coli* BL21 transformed with pET-28a-PhoP and purified with nickel-nitrilotriacetic acid (Ni-NTA) resin (see Fig. S1 in the supplemental material). DAP-seq was performed with phosphorylated recombinant His$_6$-PhoP incubated with the sheared genome of RA-YM in two duplicates, and the high-throughput data were generated from the PhoP-binding DNA. The DAP-seq analysis of phosphorylated His$_6$-PhoP identified 583 enriched peaks covering the upstream regions of 764 genes, compared with the negative control performed without His$_6$-PhoP. These peaks were distributed along the *R. anatipestifer* genome with no apparent pattern (Fig. 1). The MEME suite tools were used for probing for the overrepresented sequences near the center of 583 screened peaks (24), and a conserved motif was generated around 579 peaks (Fig. 2B), while absent in the promoter region of *KYF39_00705*, *KYF39_01335* (*uvrA*), *KYF39_01340*, and *KYF39_01345* (*rpsA*). Remarkably, as determined by comparison to the prokaryote motif database, the motif enriched from PhoP is highly similar to the motif of ArcA from *E. coli* K-12 (Fig. 2A), which regulates a wide variety of aerobic enzymes under anaerobic conditions (25–28). The ability to survive oxidative stress might contribute to the virulence of pathogens, especially anaerobic bacteria, because the bacteria must be capable of surviving the oxidative stress from the host defenses, particularly in the relatively aerobic tissue of the air sacs, during an infection (29, 30).

To validate the credibility of the DAP-seq data, the binding locus upstream of the PhoPR operon was first validated by electrophoretic mobility shift assay (EMSA). For the majority of two-component systems, a self-regulation mechanism exists (31), and the DAP-seq data were found to generate a distinct peak upstream of *phoPR*. Consistent with the DAP-seq results and predicted motif, the EMSA demonstrated PhoP to bind a 35-nucleotide (nt) region in the region from −156 to −121 upstream of *phoPR* (Fig. 3). In addition, we sought to investigate whether the DNA-binding domain of PhoP could bind the target DNA alone *in vitro*. The EMSA results indicated that the DNA-binding domain (DBD) of PhoP is not sufficient for the DNA binding function (Fig. 3B). Subsequently, the band shifts were observed with the promoter regions of seven other annotated candidate genes (*moxR*, *KYF39_00915*, *KYF39_00905*, *KYF39_07405*, *KYF39_06865*, *dnaG*, and *dedA*) (Fig. 4). In other research on TCS, phosphorylation has been

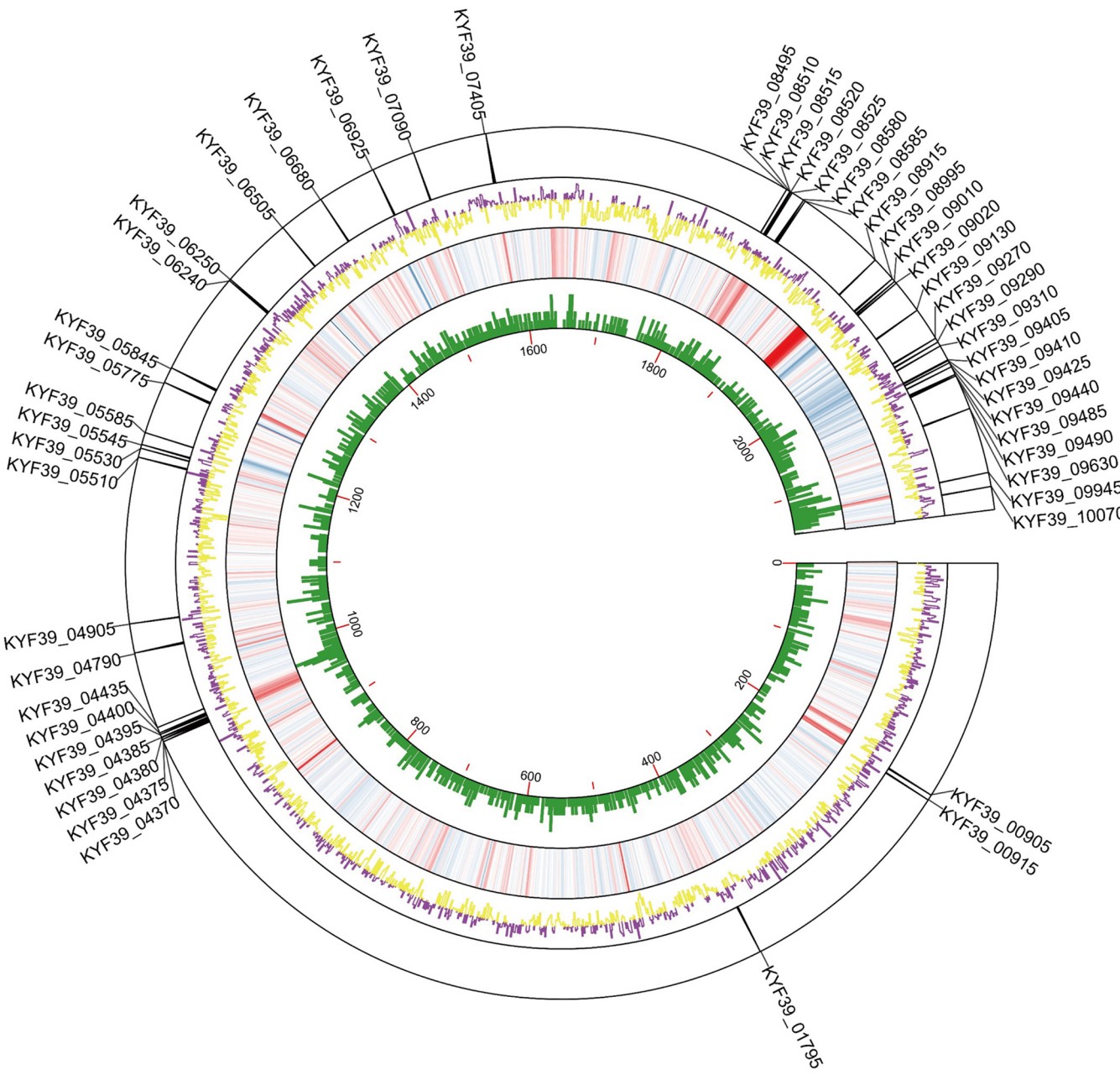

**FIG 1** Genome-wide overview of the data generated by DAP-seq and RNA-seq. The DAP-seq analysis reveals the enrichment of PhoP binding sites throughout the *Riemerella anatipestifer* RA-YM genome. The purple and yellow tracks represent the read coverage of PhoP-bound sequences throughout the *R. anatipestifer* RA-YM chromosome, shown as counts per million reads. The regions of the PhoP-binding DNA produced 583 peaks covering the upstream regions of 764 genes. The heat map track represents the RNA-seq data by $\log_2$ fold change of the expression when the $\Delta phoP$ strain was compared to RA-YM. The green track shows the gene density in RA-YM. The outmost layer of the circle shows the PhoP direct regulons. The graph was visualized by TBtools.

reported to stimulate the DNA binding ability of RR, but previous reports have also shown that phosphorylation does not affect DNA binding activity *in vitro* (32). Therefore, the unphosphorylated treatment group was added to the EMSA when validating the binding regions upstream of *moxR*, *dnaG*, *KYF39_00915*, and *KYF39_00905* (Fig. 4A to D). The dissociation constant ($K_d$), i.e., the protein concentration that resulted in 50% DNA bound with PhoP, was calculated (Fig. 4C and D). PhoP bound the upstream region of *KYF39_00915* with a $K_d$ value of 29.81 nM, while the phosphorylated PhoP had a $K_d$ value of 33.87 nM. The $K_d$ values were 39.19 nM and 39.55 nM (phosphorylated) for binding with the upstream region of *KYF39_00905*. The results indicated that the unphosphorylated PhoP of

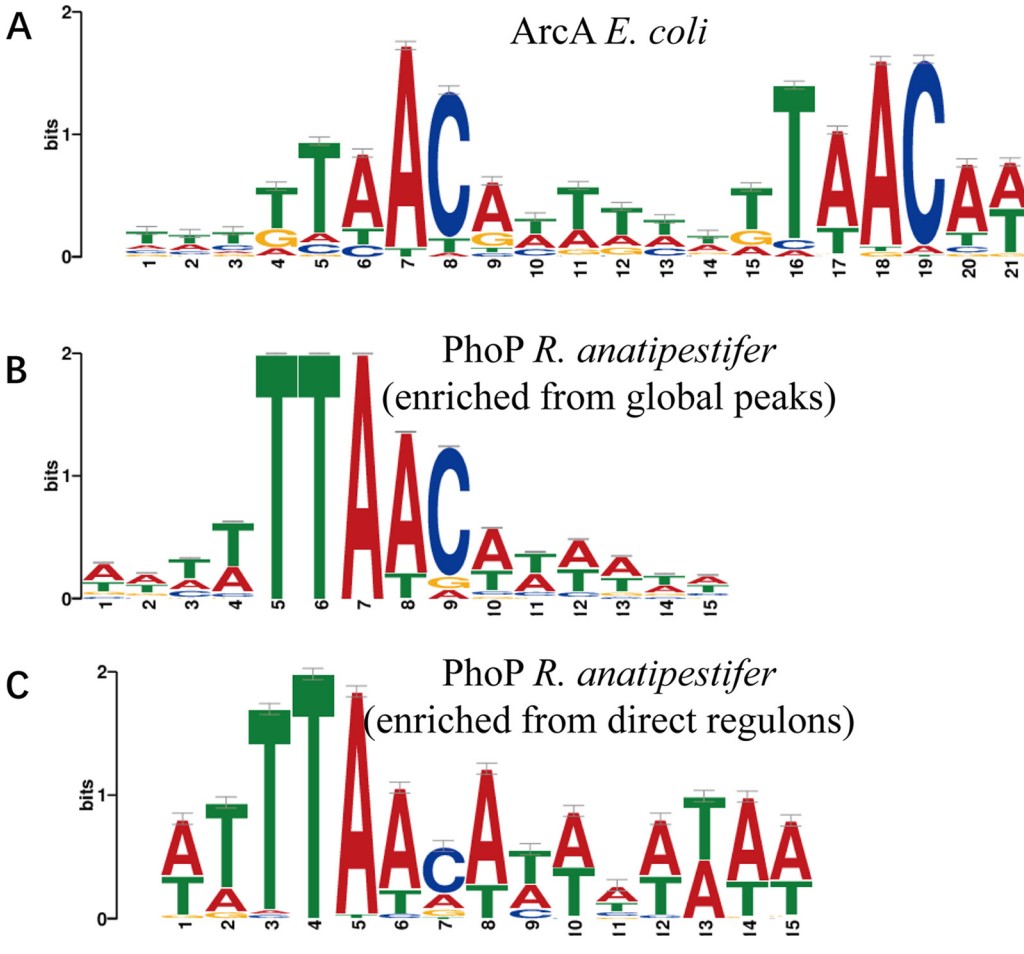

**FIG 2** The most enriched motifs for the PhoP DAP-seq data set. (A) DNA-binding motif of ArcA in *Escherichia coli* K-12 substrain MG1655. (B) DNA binding motif enriched from 583 peaks that cover the upstream regions of 764 genes in RA-YM. (C) DNA binding motif enriched from the peaks covering the upstream regions of 50 candidate PhoP-binding genes that are differentially expressed in the Δ*phoP* strain. The most conserved nucleotides are TTTAACA. The motifs were generated using MEME (https://meme-suite.org/meme/).

*R. anatipestifer* could bind DNA *in vitro*, while the phosphorylation of *phoP* was confirmed by Phos-tag SDS-PAGE (Fig. S2) (33).

**Transcriptome characterization of PhoR and PhoP in *R. anatipestifer*.** We constructed a *phoPR* double-deletion strain in our previous research and analyzed its related phenotypic characteristics and transcriptome (17). However, due to the incompleteness of the reference genome in the previous version, the reference genome of RA-YM was updated with PacBio sequencing as a reference genome, and the transcriptome data of the Δ*phoPR* double-deletion strain were reprocessed. Subsequently, to further demonstrate the regulation mechanism of PhoPR in *R. anatipestifer*, strains with single deletions of *phoR* and *phoP* were constructed via homologous recombination (Fig. S3), and an integrated characterization of their regulons was further performed via RNA-seq. A total of 183 differentially expressed genes (DEGs) were identified in the Δ*phoP* strain compared with the wild type (WT), including 123 downregulated genes and 60 upregulated genes (Fig. 1 and 5A and C). The expression of 136 genes in the Δ*phoR* strain was altered compared to that in the WT; 45 were significantly downregulated and 91 were significantly upregulated (Fig. 5B and C). Independent validation of the RNA-seq data for a subset of genes obtained using quantitative reverse transcription-PCR (RT-qPCR) confirmed the accuracy of the high-throughput results (Fig. S4). A comparison of the genes regulated by PhoR and PhoP indicated that 59 genes were differentially expressed in both the mutant strains (Table S3). Further analysis of these 59 genes

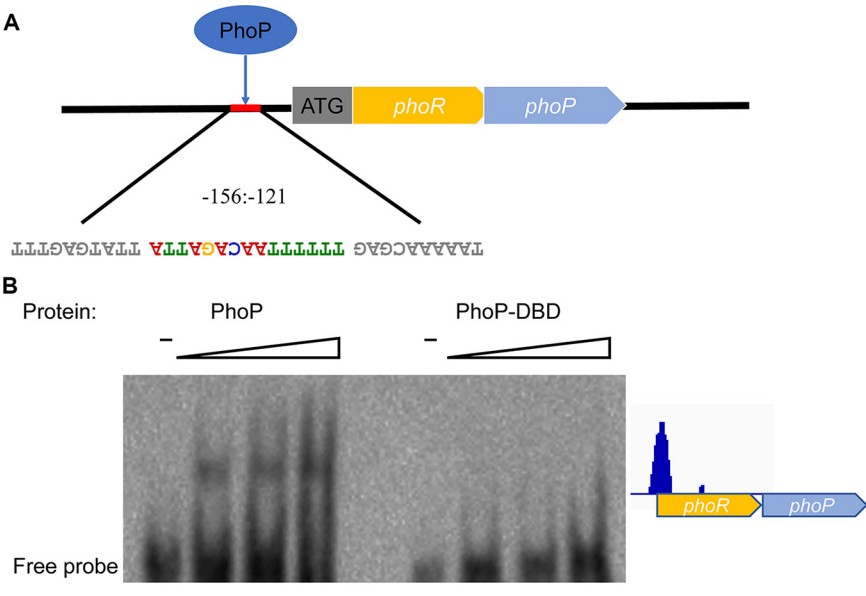

**FIG 3** Determination of PhoP self-regulation. (A) PhoP binds the upstream regions of its operon to self-regulate. (B) The purified PhoP and PhoP-DBD were used for EMSAs with the target DNA from the promoter region of the PhoPR TCS. The predicted PhoP binding site is noted below the EMSA results, and the conserved motif is highlighted in color.

showed that the expression of 57 genes changed according to the same trend, including 35 genes downregulated and 22 genes upregulated, in the Δ*phoP* and Δ*phoR* strains (Fig. 5D), and only two genes displayed different trends in regulation. One of the two genes is *phoP* itself, and the other is *KYF39_06325*, which is annotated as a hypothetical protein (Table S3).

**Integration of transcriptome profiling with PhoP binding sites.** Furthermore, the candidate target genes that are directly regulated by PhoP were identified by combining data from the DAP-seq of PhoP and RNA-seq of the Δ*phoP* mutant (Fig. 6). From the RNA-seq of the Δ*phoP* mutant, we can know the genes directly or indirectly regulated by PhoP, and the DAP-seq of PhoP provides information on PhoP-binding sites on the genome. The differential expression of 764 genes from DAP-seq was analyzed. Surprisingly, only 50 candidate target genes which had the predicted motif located in the upstream regions were differentially expressed by the Δ*phoP* strain relative to the WT, including 34 upregulated genes and 16 downregulated (Tables 1 and 2). Forty-one genes with annotation were located in the membrane. We noticed that there were several candidate genes related to aerobic tolerance or anaerobic respiration metabolism: two cytochrome *c* genes, a NosL gene, and a gene encoding an electron transfer flavoprotein subunit were upregulated, while the *Bacteroides* aerotolerance (Bat) operon was downregulated. We further tested whether PhoP could regulate the Bat operon to respond to the $H_2O_2$. The mRNA levels of *batA* and *batC* were found to differ in different strains or different oxygen content treatments. The genes *batA* and *batC* were upregulated when $H_2O_2$ was added and downregulated either when treated anaerobically; when *phoP* or *phoR* was deleted, the survival rate of the Δ*phoP* and Δ*phoR* strains decreased compared with that of the WT after exposure to 10 mM $H_2O_2$ (Fig. S5). The Bat operon was speculated to play a role in resisting oxidative stress in *R. anatipestifer*, and the Bat operon was directly regulated by PhoP (Fig. S6).

In addition, we sought to identify whether a distinctive motif from the peaks in the upstream region of 50 direct regulons is different from the one enriched from the global DAP-seq data. The motif generated from peaks of 50 direct regulons is approximately the same as that of 583 genes (Fig. 2C), but there might be differences in the significance of the motif due to the decrease in the sample number.

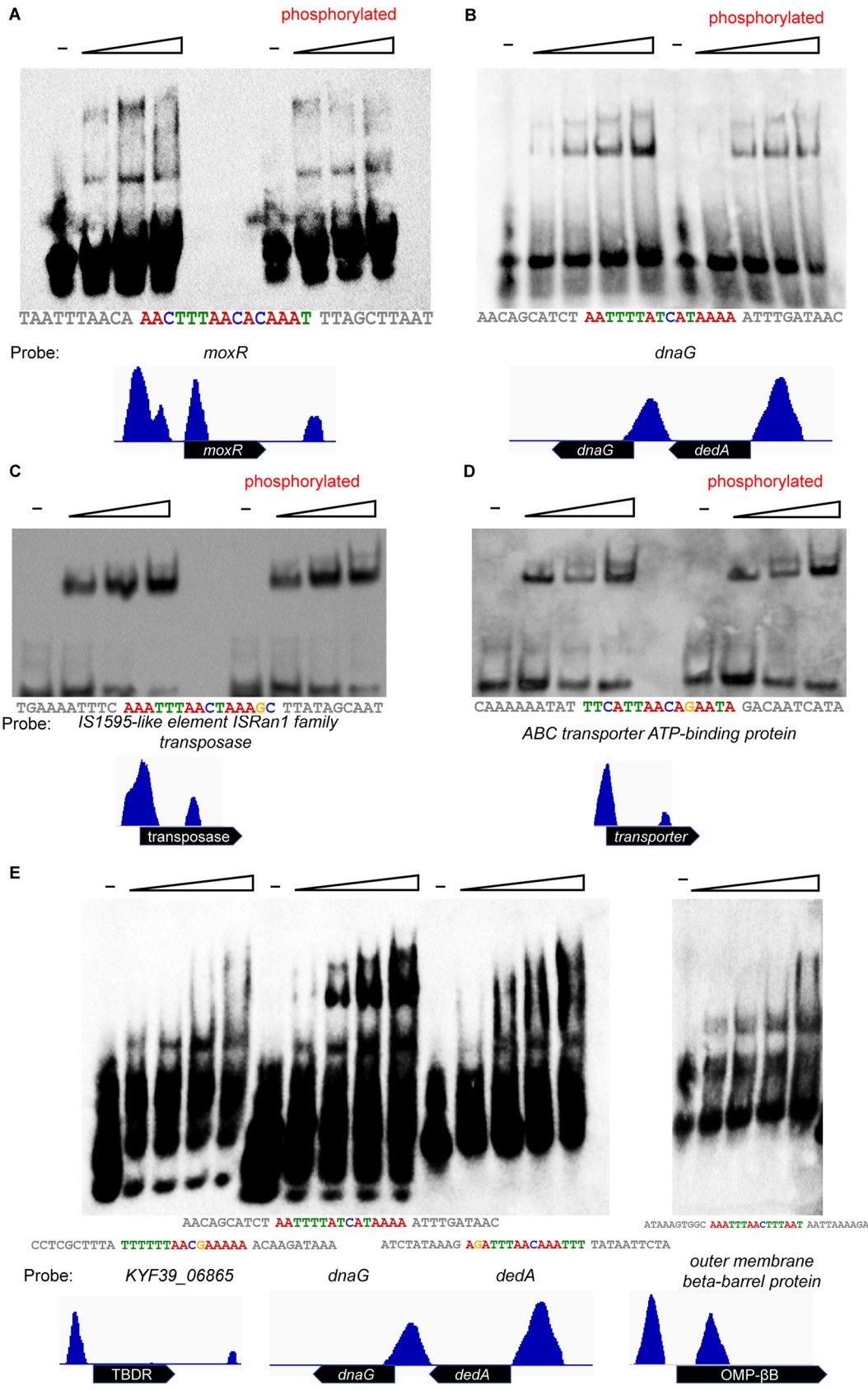

**FIG 4** Identification of PhoP binding to the upstream regions of selected genes via EMSA. (A to D) Purified PhoP (phosphorylated by acetyl phosphate or not) was used for EMSAs with the target DNA from the promoter regions of *moxR*, *dnaG*, *KYF39_00915*

Subsequently, the transcriptome of the Δ*phoR* mutant was also analyzed, and the Bat operon was noticed to be significantly downregulated in the absence of *phoR*. Moreover, only 4/16 of candidate genes directly downregulated in the Δ*phoP* mutant were not among the DEGs of the Δ*phoR* mutant, but the number of candidate genes that were directly upregulated in the Δ*phoP* mutant was found to be greatly reduced in the Δ*phoR* mutant. The results suggested a certain degree of inconsistency between the regulation of *phoP* and *phoR* to the downstream regulons, which might be due to the cross talk of the phosphorylation signal transduction in *R. anatipestifer* (34).

**Contribution of PhoR and PhoP to pathogenicity in a duck model.** Since there were differences between the DEGs from the RNA-seq of Δ*phoR* and Δ*phoP* single-deletion strains and the previously constructed Δ*phoPR* strain, we wondered whether the Δ*phoR* and Δ*phoP* strains share a similar phenotype of pathogenicity in a duck model with the Δ*phoPR* strain. Briefly, 7-day-old ducklings were infected with a range of doses ($1 \times 10^5$ to $1 \times 10^9$ CFU per duckling) of Δ*phoR* and Δ*phoP* strains via injection through flippers, and the median lethal dose (LD$_{50}$) was assessed. The ducklings injected with the WT strain began to die on the first day, and the number of deaths continued to increase over the following 5 days. The surviving ducklings returned to a normal diet after 7 days of bradykinesia. The ducklings infected with the Δ*phoR* strain displayed good mental conditions, normal diet, and normal behavior without any adverse effects. The ducklings infected with the Δ*phoP* strain showed lethargy after injection, preferred huddling together than moving, and gradually returned to a normal diet and behavior after the 4th day. According to the modified Kirschner method (35), the median lethal doses of WT and deletion strains were calculated. The results are shown in Table S4. The LD$_{50}$s of RA-YM and the Δ*phoP* and Δ*phoR* strains were $3.98 \times 10^4$ CFU/mL, $4.22 \times 10^9$ CFU/mL, and $1.88 \times 10^9$ CFU/mL, respectively. The virulence of the Δ*phoP* strain was $4.72 \times 10^4$ times lower than that of RA-YM, while that of the Δ*phoR* strain was $1.06 \times 10^5$ times lower. The results indicated that *phoP* and *phoR* are the essential virulence-related genes of *R. anatipestifer*.

To assess the impact of *phoR* and *phoP* on the *R. anatipestifer* burden during infection, blood and tissues were isolated at 24 and 48 h after infection, homogenized, and then plated to enumerate CFU. There were significantly fewer bacteria observed in the blood and tissues of ducks infected with the Δ*phoP* or Δ*phoR* strain (Fig. 7), implying that *phoR* and *phoP* are essential in the virulence of *R. anatipestifer*.

At 24 h and 48 h after injection, the liver, heart, spleen, and brain tissues from ducklings treated with PBS or infected with RA-YM or the Δ*phoP* or Δ*phoR* strain were fixed in formalin and subjected to histopathologic examinations. The duckling tissues showed significant pathological changes when infected with RA-YM at 24 h and 48 h. The abnormality was observed in the structures of the myocardial, liver, and spleen tissues, with necrosis and degeneration apparent except in the brain (Fig. 8, black arrows). Some myocardial fibers were disarranged, and many inflammatory cells infiltrated the myocardial space, as shown by the red arrows (Fig. 8). The hepatocytes were found to undergo extensive fatty degeneration, as shown by the red arrows in live tissue (Fig. 8). The structure of the spleen tissue was abnormal; some lymphocytes were necrotic and degenerative, and the nucleus was fragmented and pyknotic. The structure of brain tissue was also abnormal; some neurons were hyperchromatic and underwent pyknosis, and there was an evident phenomenon of psychrophilic cells. The glial cells could be seen to undergo phagocytosis (Fig. 8, green arrows). The tissues from mock-infected ducklings and ducklings infected with Δ*phoP* and Δ*phoR* strains did not show obvious pathological changes. The results showed that the pathological changes in the tissues and organs of the ducklings were significantly reduced after 24 h and 48

**FIG 4** Legend (Continued)

(ISRan1 family transposase), and *KYF39_00905* (ABC transporter). The predicted PhoP binding site is noted below the EMSA results, and the conserved motifs are highlighted in color. (E) Purified PhoP was used for EMSAs with the target DNA from the promoter regions of *KYF39_06865* (TonB-dependent receptor), *dnaG*, *dedA*, and *KYF39_07405* (outer membrane beta-barrel protein). The predicted PhoP binding sites are noted below the EMSA results, and conserved motifs are highlighted in color.

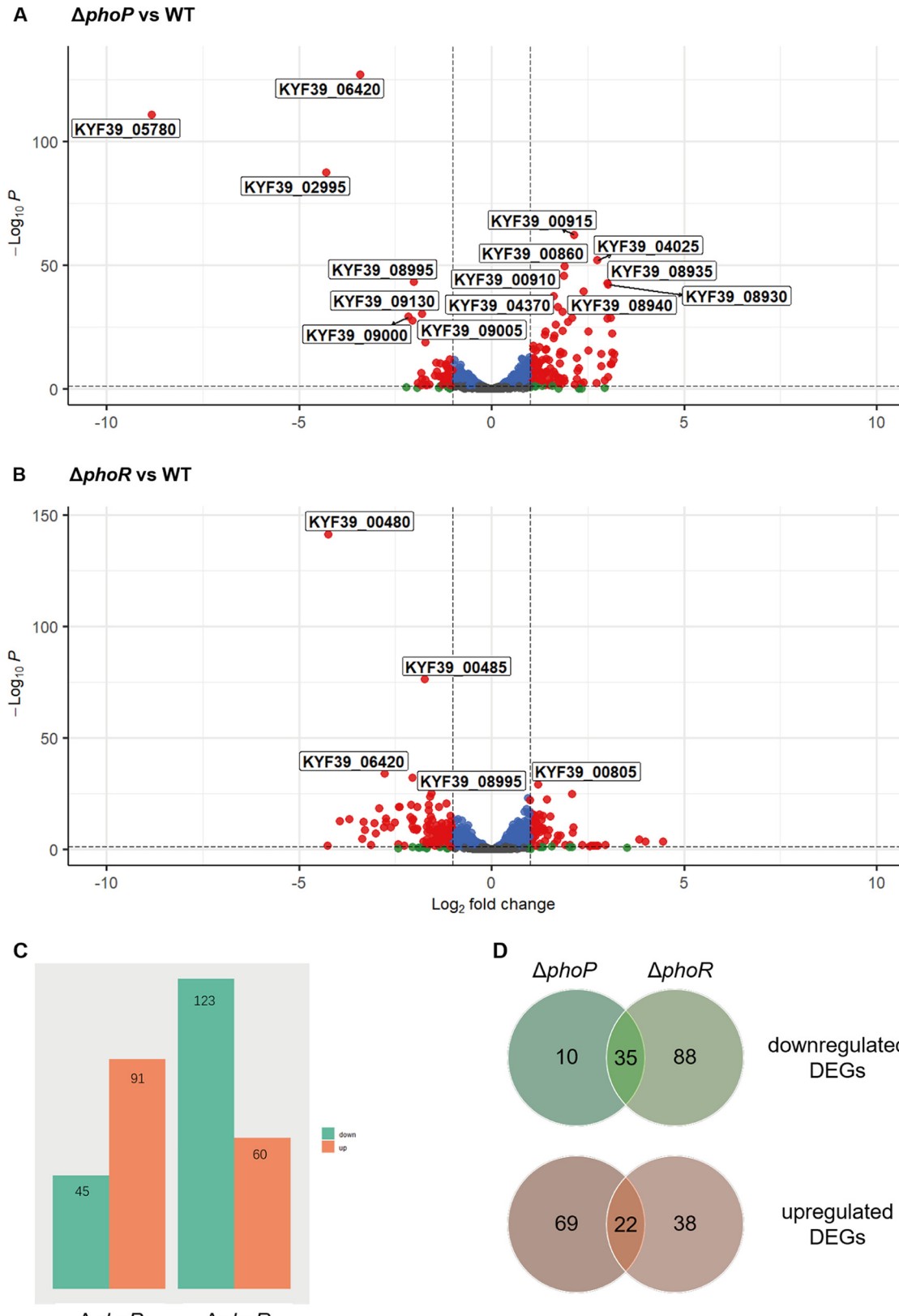

**FIG 5** Differentially expressed genes of the Δ*phoR* and Δ*phoP* mutant strains compared to the WT. (A and B) Volcano plot showing differential expression of genes in the Δ*phoR* and Δ*phoP* strains, respectively. DEGs (absolute value of log$_2$ FC > 1; false discovery rate < 0.05) are highlighted (red). (C) Bar chart showing the number of genes whose normalized usage was significantly (adjusted *P* value < 0.01) reduced (aqua) or enhanced (coral) over 2-fold in the Δ*phoP* and Δ*phoR* strains, respectively. (D) Venn diagram showing the overlapping genes that were downregulated (green system) and upregulated (orange system) in the Δ*phoP* and Δ*phoR* strains.

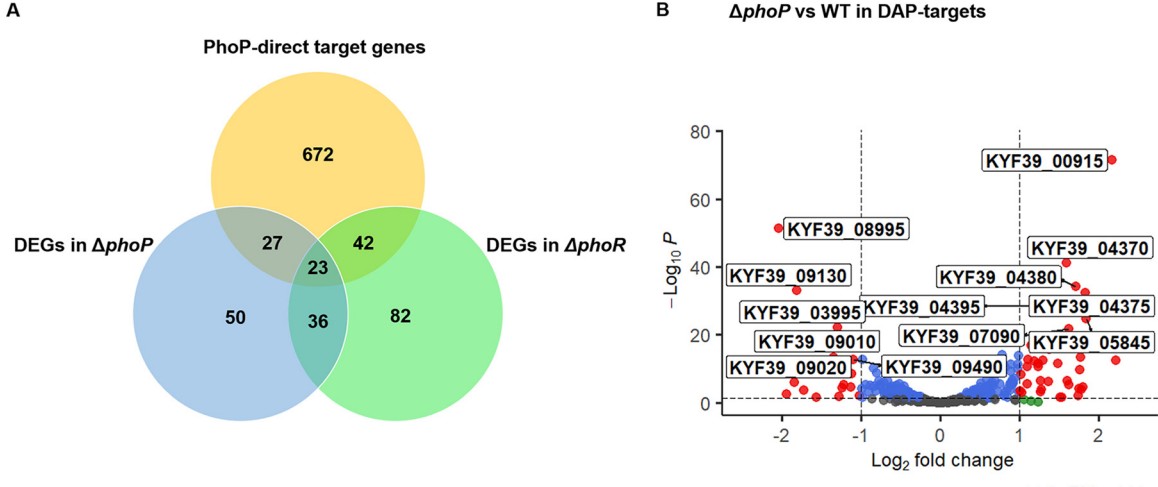

**FIG 6** Combination analysis of DAP-seq and RNA-seq. (A) Venn diagram showing the overlapping genes between the DAP-seq and RNA-seq data. The PhoP-direct target genes from DAP-seq are in the orange section; DEGs in the Δ*phoR* and Δ*phoP* strains identified from RNA-seq are in green and blue, respectively. (B) Volcano plot showing the differential expression of the 764 genes of the PhoP targets. DEGs (absolute value of $\log_2$ FC > 1; false discovery rate < 0.05) are indicated in red.

h of infection with the Δ*phoP* and Δ*phoR* strains. In conclusion, the Δ*phoP* and Δ*phoR* mutants almost lost its pathogenicity to ducklings. Thus, PhoP and PhoR were proven to be essential for the pathogenicity of *R. anatipestifer*.

## DISCUSSION

The PhoPR TCS is essential for pathogenicity in *R. anatipestifer*. Here, the regulatory mechanism of PhoP in *R. anatipestifer* was explored by DAP-seq and RNA-seq. This study provides new insight into how PhoPR regulates the transcriptome accordingly and established a new model for studying transcriptional regulation. To identify the direct binding sites of PhoP, DAP-seq was utilized to get a global view of the regulons. To study the directly regulated genes, RNA-seq was used to study the regulons of *phoP* after constructing the Δ*phoP* and Δ*phoR* strains. Finally, the direct regulons of PhoP were found, and *phoP* and *phoR* were identified as essential virulence-related factors for *R. anatipestifer*.

The PhoPR two-component system exists in most Gram-positive bacteria, and PhoBR exists in most Gram-negative bacteria. The PhoPR and PhoBR TCSs are involved in regulating phosphate homeostasis. The expression of these TCSs depends on the concentration of phosphate in the environment and is upregulated to regulate the downstream-related genes under low-phosphate conditions, *phoP* and *phoB* are collectively called *pho* regulators (34). In addition to regulating phosphate homeostasis, PhoPR is also believed to regulate other cellular processes such as biofilm formation, cell wall metabolism, vitamin metabolism, and bacterial respiration (31). Most importantly, PhoPR has also been shown to regulate the pathogenesis of many pathogens. The annotated homologs of PhoP and PhoR in *Bacillus anthracis* were significantly similar to the PhoPR proteins of other pathogens (36), and the PhoPR TCS has also been reported in *Bacillus subtilis* (37), *Streptomyces* (38), and *M. tuberculosis* (39).

In many bacteria, such as *B. subtilis*, *Salmonella enterica* serovar Typhi, and *M. tuberculosis*, *phoP* belongs to the *pho* regulators as a self-regulator (31). The self-regulation of PhoP in *R. anatipestifer* was first identified via EMSA, and the binding was located in the region from 156 to 121 upstream of the start codon ATG of *phoPR* (Fig. 3). PhoP is known to regulate the virulence-related genes in several pathogens, such as *Enterococcus faecalis*, *Corynebacterium pseudotuberculosis*, *M. tuberculosis*, and *E. coli* (40–43). The PhoPR two-component system is very important for the virulence of *M. tuberculosis* and can be used as a potential target for developing antituberculosis therapy

**TABLE 1** DEGs directly downregulated/repressed by PhoP

| Gene | Log$_2$ fold change | P value | Annotation |
|---|---|---|---|
| KYF39_00905 | 3.12 | 1.39E−15 | ABC transporter ATP-binding protein |
| KYF39_00915 | 2.14 | 2.00E−89 | IS1595-like element ISRan1 family |
| KYF39_01795 | 1.65 | 6.17E−04 | Hypothetical protein |
| KYF39_04355 | 1.53 | 9.70E−03 | tRNA-Glu |
| KYF39_04370 | 1.61 | 5.32E−32 | *N*-Acetylmuramoyl-L-alanine amidase |
| KYF39_04375 | 1.84 | 1.27E−18 | LPS assembly protein LptD |
| KYF39_04380 | 1.72 | 2.97E−37 | RidA family protein |
| KYF39_04385 | 1.48 | 3.26E−12 | Hypothetical protein |
| KYF39_04395 | 1.67 | 2.29E−28 | Citrate synthase |
| KYF39_04400 | 1.16 | 8.51E−12 | AhpC/TSA family protein |
| KYF39_04435 | 1.01 | 3.72E−07 | Type B 50S ribosomal protein L36 |
| KYF39_04790 | 1.25 | 2.90E−07 | Do family serine endopeptidase |
| KYF39_04905 | 1.50 | 1.47E−02 | Hypothetical protein |
| KYF39_05510 | 1.78 | 1.12E−08 | Adenosine deaminase |
| KYF39_05530 | 1.11 | 2.81E−16 | Hypothetical protein |
| KYF39_05545 | 1.01 | 2.53E−05 | IS982-like element ISRa1 family transposase |
| KYF39_05775 | 1.09 | 1.18E−08 | HAMP domain-containing histidine kinase |
| KYF39_05845 | 1.84 | 5.22E−04 | ABC transporter ATP-binding protein/permease |
| KYF39_06240 | 1.29 | 3.06E−05 | Hypothetical protein |
| KYF39_06250 | 1.77 | 1.46E−11 | Hypothetical protein |
| KYF39_06505 | 1.25 | 4.09E−11 | Hypothetical protein |
| KYF39_06680 | 1.39 | 1.13E−18 | GLPGLI family protein |
| KYF39_06925 | 1.38 | 2.58E−12 | DUF4407 domain-containing protein |
| KYF39_07090 | 1.63 | 4.55E−30 | GLPGLI family protein |
| KYF39_07405 | 2.22 | 4.55E−10 | Outer membrane beta-barrel protein |
| KYF39_08495 | 1.88 | 9.91E−07 | Nitrous oxide reductase accessory protein NosL |
| KYF39_08510 | 1.32 | 9.53E−05 | Cytochrome *c* |
| KYF39_08515 | 1.76 | 1.44E−07 | *c*-type cytochrome |
| KYF39_08520 | 1.79 | 3.09E−04 | Hypothetical protein |
| KYF39_08525 | 1.74 | 3.60E−04 | Hypothetical protein |
| KYF39_08580 | 2.85 | 1.27E−02 | TonB-dependent receptor |
| KYF39_08585 | 1.10 | 1.04E−06 | Alpha/beta hydrolase |
| KYF39_08915 | 3.12 | 3.27E−03 | Hypothetical protein |
| KYF39_09630 | 1.21 | 4.55E−07 | Electron transfer flavoprotein subunit alpha/FixB family protein |

(44). Humans do not possess proteins of these two-component systems, so they are suitable targets for antibacterial drugs. Previous studies have shown that the biological characteristics of the Δ*phoPR* strain are the same as those of RA-YM, and the Δ*phoPR* strain's virulence and its ability to damage the host tissues and organs are significantly

**TABLE 2** DEGs directly upregulated by PhoP

| Gene | Log$_2$ fold change | P value | Annotation |
|---|---|---|---|
| KYF39_05585 | −1.02 | 2.19E−03 | Hypothetical protein |
| KYF39_08995 | −2.03 | 4.22E−35 | AAA family ATPase (MoxR) |
| KYF39_09010 | −1.33 | 4.32E−08 | VWA domain-containing protein (BatA) |
| KYF39_09020 | −1.44 | 6.99E−12 | Tetratricopeptide repeat protein (BatC) |
| KYF39_09130 | −1.81 | 6.38E−34 | GLPGLI family protein |
| KYF39_09270 | −1.83 | 6.41E−08 | GLPGLI family protein |
| KYF39_09290 | −1.92 | 3.67E−07 | Hypothetical protein |
| KYF39_09310 | −1.22 | 2.57E−06 | HXXEE domain-containing protein |
| KYF39_09405 | −1.12 | 1.24E−10 | Hypothetical protein |
| KYF39_09410 | −1.72 | 4.30E−04 | Hypothetical protein |
| KYF39_09425 | −1.29 | 1.07E−05 | Hypothetical protein |
| KYF39_09440 | −1.13 | 1.90E−04 | GLPGLI family protein |
| KYF39_09485 | −1.19 | 2.97E−12 | DNA primase (DnaG) |
| KYF39_09490 | −1.10 | 1.46E−19 | VTT domain-containing |
| KYF39_09945 | −1.27 | 9.18E−03 | Hypothetical protein |
| KYF39_10070 | −1.07 | 6.08E−03 | Hypothetical protein |

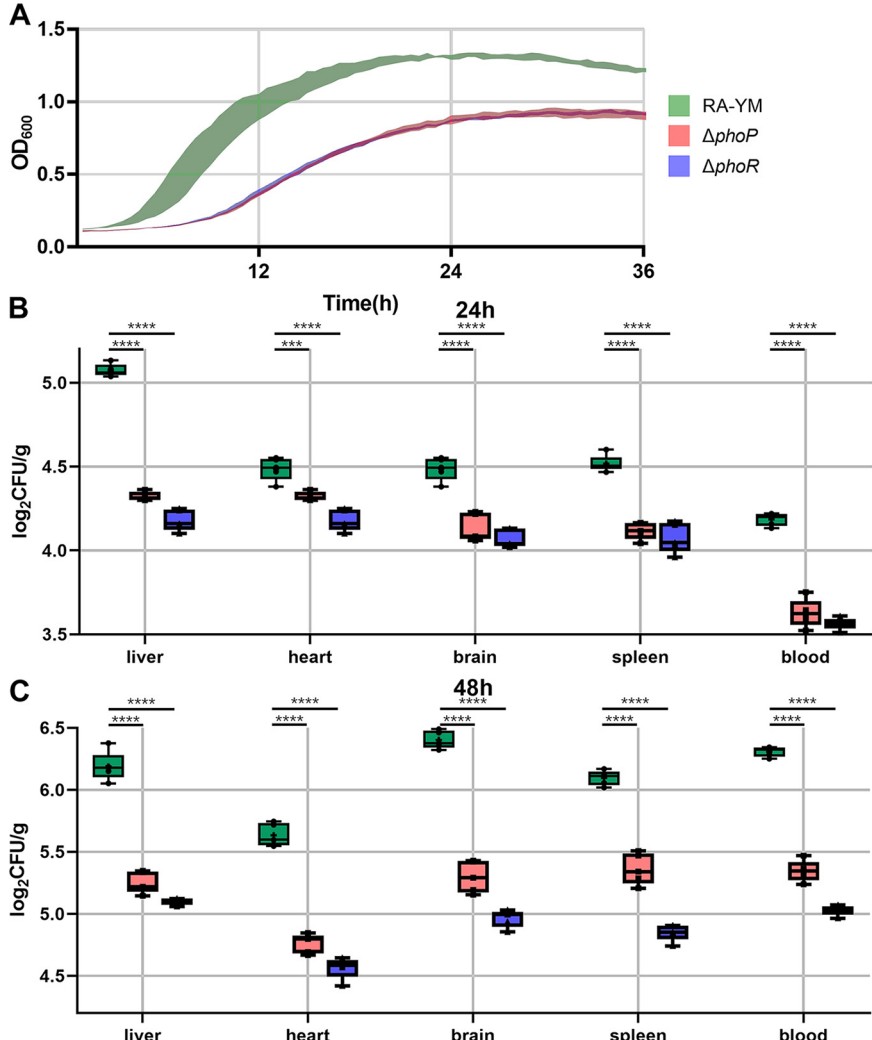

**FIG 7** Biological characteristics of the RA-YM, Δ*phoP*, and Δ*phoR* strains. (A) Growth curves of Δ*phoP*, Δ*phoR*, and RA-YM strains in TSB. The indicated strains were grown to the exponential phase ($OD_{600}$ = 0.6 to 0.8) in TSB, at which point they were harvested by centrifugation, resuspended to an $OD_{600}$ of 1 in TSB, and then transferred to fresh TSB medium at a dilution of 1:100. $OD_{600}$ was measured every 30 min, and five repetitions were carried out for each strain. Standard deviations are depicted with colored shading. (B and C) Δ*phoP*, Δ*phoR*, and RA-YM strains obtained from blood and tissues of infected ducks after 24 h and 48 h, respectively. Two-week-old ducklings were challenged with $10^5$ CFU of the Δ*phoP* mutant, the Δ*phoR* mutant, and RA-YM. After 24 or 48 h, the blood and tissues were harvested, and the numbers of CFU were determined by serial dilution and pour plating. ***, $P < 0.0002$; ****, $P < 0.0001$.

lower, with an $LD_{50}$ of $>10^{10}$ CFU, and hence, this mutant could be considered an avirulent strain (17).

To further explore the regulation of PhoPR in *R. anatipestifer*, DAP-seq was applied to investigate the PhoP-binding sites around the genome globally. DAP-seq allows us to use a phosphoryl group donor to mimic phosphorylation of RR, avoiding the limitation of unknown activating signals of PhoP. DAP-seq does not require a specific antibody or tagged transgenic lines, unlike chromatin immunoprecipitation sequencing (ChIP-seq), and only requires an affinity method for the tag-labeled response regulator. However, there is also a limitation to DAP-seq. As a method to study protein-DNA interaction *in vitro*, the binding sequence produced by DAP-seq is comparable to the direct binding sequence of the protein itself, but it cannot completely equal the binding ability or site of RR *in vivo*, and RR *in vivo* may have some indirect binding ability or cross talk (45).

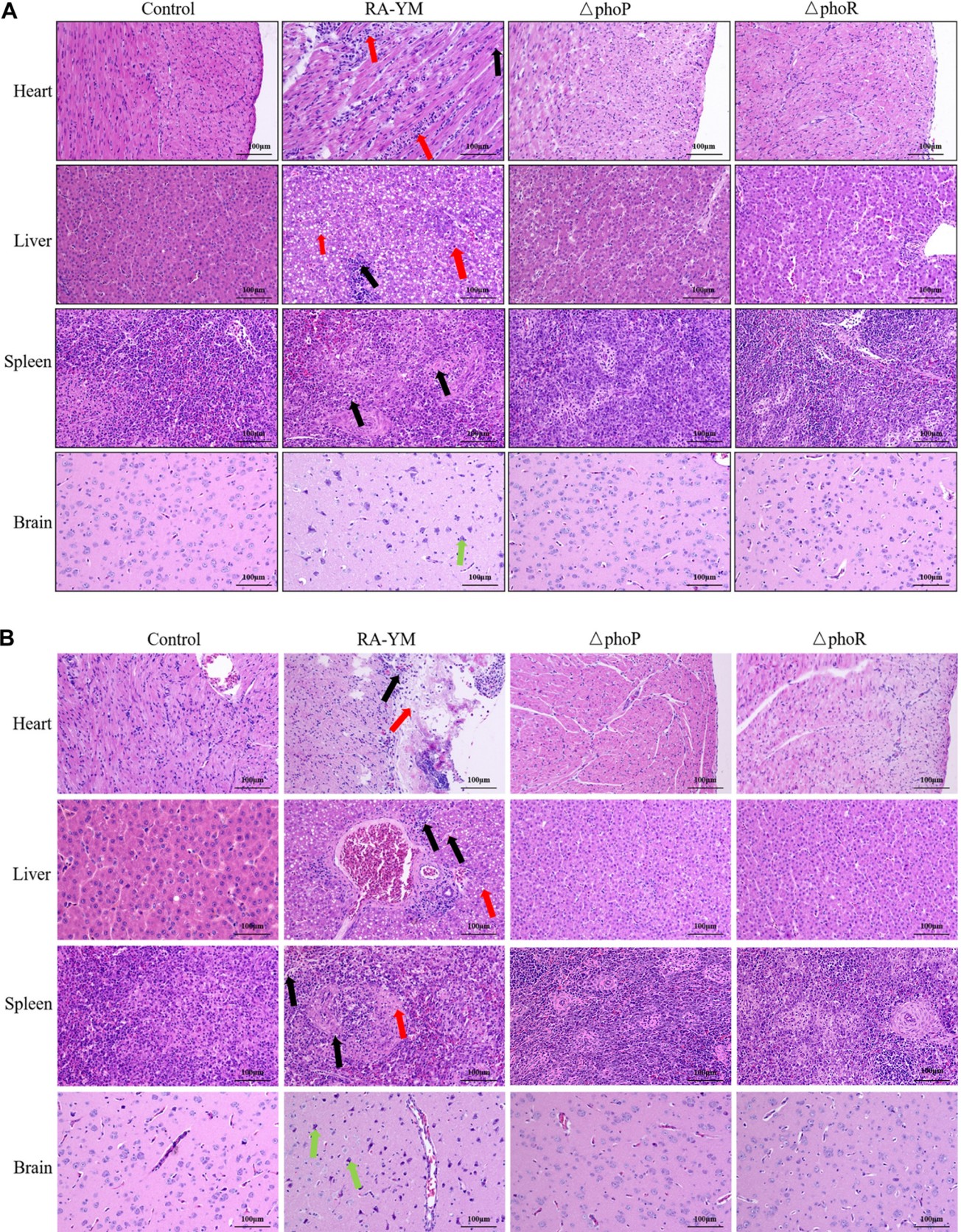

**FIG 8** Histopathological analysis of the duck tissues infected with RA-YM and the Δ*phoP* and Δ*phoR* strains. (A and B) Heart, liver, spleen, and brain samples were collected from ducks infected with RA-YM and the Δ*phoP* and Δ*phoR* mutants at 24 h (A) and 48 h (B). Black arrows indicate necrosis and degeneration in the tissues. Red arrows indicate fatty degenerated hepatocytes in the hepatic tissues and disarranged myocardial fibers and the inflammatory cells which infiltrated the myocardial space in the heart. Green arrows indicate phagocytosis surrounded by glial cells in the brain.

Of the peaks generated from DAP-seq data, 8 (Fig. 3 and 4) were identified via EMSA. The DNA-binding domain (DBD) of PhoP was also investigated to find out whether it could bind the peaks independently, and the EMSA results indicated the necessity of an N-terminal receiver domain (RD) for the DNA-binding ability of PhoP in *R. anatipestifer* (Fig. 3B). For NarL in *E. coli*, the DBD was inhibited by the N-terminal RD in a process in which the C-terminal DBD folds back into the N-terminal RD, leading to the loss of the DNA-binding activity (46). In this case, phosphorylation is necessary for releasing the inhibition, allowing RD dimerization or oligomerization and allowing DBD to bind to the target DNA. OmpR binds to DNA as a monomer with high affinity resulting from the stimulated phosphorylation and subsequent dimerization by RD, while OmpR-DBD binds to DNA with a low affinity incapable of transcriptional initiation (47, 48). However, in *B. subtilis*, the opposite phenomenon was observed: the N terminus and C terminus of PhoP were found to function independently and did not mutually inhibit their domain functions (49).

We also wondered whether phosphorylation is essential for PhoP binding to the DNA or enhancing the DNA-binding ability *in vitro*. The EMSA results showed that phosphorylation is not necessary for the DNA binding *in vitro* and did not significantly enhance the DNA binding (Fig. 4A to D). Similar situations were observed in the DNA binding of ComE in *Streptococcus mutans* (32) and PhoP in *B. subtilis* (49). Increasing concentrations of acetyl-P or phosphoramidite were not significant for the DNA binding ability of ComE, and there was a diminution of the extent of the shift band that appeared in EMSA which was instead probably due to the RR being already been phosphorylated endogenously during the purification from *E. coli* BL21(DE3). Spo0A is heterologously phosphorylated and functional in DNA-binding when expressed in *E. coli* (50). Therefore, Phos-tag SDS-PAGE was utilized to investigate whether PhoP could be phosphorylated by acetyl-P *in vitro*, and the results showed that PhoP could be phosphorylated *in vitro* with acetyl-P in a time-dependent manner (Fig. S2). Although the phosphorylated PhoP was observed as a shifted band on the Phos-tag SDS-PAGE followed by Coomassie brilliant blue (CBB) staining, the degree of shifting was not like the result after dimerization or oligomerization of proteins. Our analysis of PhoP binding suggested that PhoP could bind DNA without phosphorylation *in vitro*. Future work will focus on whether phosphorylation would change the preference of the binding motif of PhoP and whether the interaction between proteins exists *in vivo*.

Furthermore, single-gene-deletion (Δ*phoP* and Δ*phoR*) strains were successfully constructed. The growth curve showed that the Δ*phoP* and Δ*phoR* mutants have decreased growth rates at every time point compared to the WT, and the cell density of WT was higher than that of the Δ*phoP* and Δ*phoR* strains in tryptic soy broth (TSB). RNA-seq was used for studying gene differential expression when *phoP* or *phoR* was deleted. The results of transcriptome analysis showed 136 genes that were differentially expressed between the Δ*phoP* strain and RA-YM, including 45 downregulated genes. There were 183 genes that were differentially expressed between the Δ*phoR* strain and RA-YM, including 123 downregulated genes. The DEGs generated from the Δ*phoP* and Δ*phoR* strains were compared, and 57 genes were differentially expressed in the same trend, including 35 DEGs that were downregulated and 22 DEGs that were upregulated when *phoR* or *phoP* was deleted.

There are some differences in the DEGs produced in the Δ*phoP*, Δ*phoR*, and Δ*phoPR* strains in previous research. From the perspective of gene or operon function, there should exist divergence in the DEGs produced by the deletion of *phoP*, *phoR* or *phoPR* due to the different regulatory pathways in cells. As a HK, PhoR not only phosphorylated the cognate RR but also possibly transferred the phosphoryl group to other RRs, while PhoP was the RR, which may also get phosphorylated by other noncognate HKs. Therefore, we believe that there is a difference in the global transcriptome between the double-deletion strain and single-deletion strains. Due to the limitations of previous research, RNA-seq of the Δ*phoPR* strain was not duplicated, and the reference genome was not appropriate, resulting in an inability to conduct statistical analysis.

Combining the data from DAP-seq and RNA-seq revealed the PhoP-binding sites in the upstream regions of 50 DEGs in the Δ*phoP* mutant. Table 1 lists the 34 directly upregulated regulons in the Δ*phoP* strain compared to the WT, including 10 hypothetical proteins, one TonB-dependent receptor (TBDR) (*KYF39_08580*), two GLPGLI family proteins (*KYF39_06680* and *KYF39_07090*), two transposases (*KYF39_00915* and *KYF39_05545*), etc. Among the 30 directly upregulated genes with annotation, the most encoded proteins were located in the membrane composed of the TBDR, ABC transporters (*KYF39_08915* and *KYF39_05845*), β-barrel outer membrane protein (*KYF39_07405*), LptD (*KYF39_04375*), NosL (*KYF39_08495*), two *c*-type cytochromes (*KYF39_08515* and *KYF39_08510*), an *N*-acetylmuramoyl-L-alanine amidase (*KYF39_04370*), a FixB family protein (*KYF39_09630*), LptD (*KYF39_04375*), and serine peptidase (*KYF39_04790*).

The membrane proteins in bacteria are related to their pathogenicity and immunogenicity; for example, the lipopolysaccharide (LPS) assembly protein LptD is responsible for assembling LPS in the outer membrane, which is an essential virulence factor in the Gram-negative bacteria (51), and can be used as a candidate for either exclusive peptide vaccines or multicomponent vaccines in *Brucella melitensis* (52). Additionally, there are two transposases in 34 regulons that are significantly directly upregulated, which might be due to the repression of transposase as a way for the cell to increase genome stability (53, 54). Table 2 lists the 16 directly downregulated regulons in the Δ*phoP* strain compared to the WT, including 7 hypothetical proteins, DnaG (*KYF39_09485*), 3 GLPGLI family proteins (*KYF39_09270*, *KYF39_09130*, and *KYF39_09440*), and three genes (*KYF39_08995*, *KYF39_09010*, and *KYF39_09020*) with loci in a complete *Bacteroides* aerotolerance operon involved in pathogenicity and aerotolerance.

*Bacteroides fragilis* must enhance survival in aerobic sites and promote opportunistic infections that induce aerotolerance and resistance to oxidative stress as physiological adaptations to its environment (55). The Bat operon was first reported in *B. fragilis*, composed of *batA*, *batB*, *batC*, *batD*, and *batE*, and *B. fragilis* was found to lose its aerotolerance after *batD* was mutated by insertion (56). Researchers also believe that in *Porphyromonas gingivalis*, the Bat operon ensures the survival ability of the pathogenic bacteria in the early process of infection at aerobic sites and that the Bat complex was possibly involved in imparting resistance to oxidative stress (57). Dieppedale et al. determined that the Bat operon was important for stress resistance and intracellular survival of *Francisella tularensis* (58). However, the aerotolerance related to the Bat operon was studied in anaerobic bacteria before, but there was no relevant research on the facultatively anaerobic bacteria or aerobic bacteria. Additionally, ArcA in *E. coli* is involved in regulation under anaerobic conditions, generating a motif similar to the motif enriched from the DAP-seq data of PhoP in *R. anatipestifer* (26).

Based on the similarity between the motif and regulatory function, *R. anatipestifer* was hypothesized to possibly adapt to the redox pressure under aerobic conditions via PhoP regulation of the Bat operon. Notably, 23 of 57 DEGs were candidate target genes of PhoP; 40% of upregulated genes have PhoP-binding sites in the upstream regions, and 12.5% of downregulated genes have PhoP-binding sites, suggesting that indirect regulation is the major mechanism of PhoP.

The effects of *phoP* and *phoR* genes on pathogenicity in ducklings were studied *in vivo*. The results showed that the $LD_{50}$ of the Δ*phoP* and Δ*phoR* mutants increased by $4.72 \times 10^4$ times and $1.06 \times 10^5$ times, respectively, compared to that of RA-YM, and the bacterial loads in tissues of ducklings infected with the Δ*phoP* and Δ*phoR* strains at 24 h and 48 h was found to be significantly lower than those in ducklings infected with RA-YM. The $LD_{50}$s and the pathogenicity in ducklings showed that *phoP* and *phoR* are essential virulence-related factors of *R. anatipestifer*. The above results showed that both *phoR* and *phoP* could affect the gene expression of *R. anatipestifer*, and the correlation between virulence and *phoR* was higher than that between virulence and *phoP*, which was consistent with the $LD_{50}$s.

The activation of the RR activity depends on its homologous HK, which has a direct interaction mechanism with RR. In previous studies, response regulators were shown

to have a major role in bacterial pathogenicity. Recently, more and more literature has reported that histidine kinase also plays a key role in bacterial pathogenicity. The MgtC virulence protein of *Salmonella enterica* serovar Enteritidis is necessary for its survival and virulence in mice (59), and this protein is conserved in several intracellular pathogens, including *Mycobacterium* (60). Although MgtC protein plays a key role in the survival in macrophages, only a few molecular targets have been identified. MgtC targets the phosphohistidine kinase and activates phosphate transport. The mutation of a single PhoR amino acid prevents binding to MgtC, which results in the loss of MgtC-mediated phosphate transport and the reduction of bacterial replication in macrophages. This suggests that the MgtC-mediated phosphate transport is necessary for *Salmonella*, but the *phoR* gene plays a major role (59). The *phoP* gene is an important virulence factor in *M. tuberculosis*, but it was recently reported that *phoR* also plays the same role (61). The phenotype of the *phoR* deletion strain of *M. tuberculosis* is similar to that of the *phoP* mutant strain, indicating that PhoP and PhoR might affect the same biochemical pathway, suggesting that PhoP and PhoR play an important role in regulating the virulence of *M. tuberculosis*.

In conclusion, both *phoR* and *phoP* are closely related to the virulence of RA-YM, and the virulence of *phoR* and *phoP* gene deletion strains in ducklings is significantly reduced. The present study used the combination of DAP-seq with RNA-seq to perform genome-wide identification of the two-component system regulon in pathogenic bacteria. The entire set of target genes under the regulation of PhoP in *R. anatipestifer* was determined, and the contributions of *phoP* and *phoR* to pathogenicity were evaluated. In addition, this study provides a more readily available method for exploring the response regulatory proteins in other two-component systems. Future work will probe the contributions of PhoP-regulated genes to virulence and aerotolerance in order to identify the molecular basis of the acute virulence defect in the Δ*phoP* mutant and the mechanism of *R. anatipestifer* resistance to reactive oxygen species (ROS). The *phoP* and *phoR* gene deletion strains can serve as candidate live vaccine strains of *R. anatipestifer* and may be applicable as ideal genetic engineering vector strains for the expression of foreign antigens.

## MATERIALS AND METHODS

**Ethics statement.** All the animal experiments were carried out in accordance with the recommendations in the Guide for the Care and Use of Laboratory Animals from Research Ethics Committee, Huazhong Agricultural University, Hubei, China. All procedures performed in studies involving animals were in accordance with the ethical standards of the institution or practice at which the studies were conducted.

**Bacterial strains and growth conditions.** All strains are listed in Table S1. RA-YM and derivates were routinely grown in tryptic soy broth (Becton, Dickinson and Company, Franklin Lakes, NJ, USA) or on tryptic soy agar (TSA) plates (Becton, Dickinson and Company, Franklin Lakes, NJ, USA) with 3% (vol/vol) supplemented newborn calf serum (NEWZERUM, Christchurch, New Zealand) at 37°C with 5% $CO_2$.

*E. coli* strain $\chi$7213 is autotrophic on diaminopimelic acid, which is used for delivering plasmid into *R. anatipestifer* via transconjugation, and was grown in lysogeny broth (LB) or LB agar supplemented with 50 $\mu$g/mL of diaminopimelic acid (Sigma-Aldrich, Darmstadt, Germany) (62). *E. coli* DH5$\alpha$ $\lambda pir$ was used for the propagation of pRE112 (63) or its derived plasmids. *E. coli* DH5$\alpha$ was used for routine cloning, and *E. coli* BL21(DE3) was used for overexpression of His$_6$-PhoP and His$_6$-PhoP-DBD. Chloramphenicol, kanamycin, and spectinomycin were used at final concentrations of 50 $\mu$g/mL, 100 $\mu$g/mL, and 100 $\mu$g/mL, respectively.

**Cloning, overexpression, and purification of His$_6$-PhoP and His$_6$-PhoP-DBD.** The recombinant plasmids pET-28a-PhoP and pET-28a-PhoP-DBD, which were used for producing His$_6$-PhoP and pET-28a-PhoP-DBD, respectively, were constructed as follows. Taking His$_6$-PhoP as an example, *phoP* was amplified by PCR using *Riemerella anatipestifer* RA-YM genomic DNA as a template with primers PhoP-F and PhoP-R (Table S2). The DNA fragment was then digested with BamHI and EcoRI and cloned into pET-28a vector digested with the same restriction enzyme. The plasmid identified by Sanger sequencing was transformed into *E. coli* BL21(DE3). *E. coli* BL21(DE3) containing pET-28a-PhoP was grown at 37°C in LB supplemented with kanamycin until an optical density at 600 nm (OD$_{600}$) of 0.4 to 0.6 was reached; then, 1 mM isopropyl-1-thio-$\beta$-D-galactopyranoside (IPTG) was added. After 8 h at 28°C, cells were harvested and then resuspended in bacterial lysis buffer (50 mM Tris-HCl, 100 mM NaCl, 1% Triton X-100, 10% glycerin; pH 8.0). After cells were crushed by a pressure cell disruptor three times followed by 15-min centrifugation at 10,000 × *g* to keep the supernatant, the unbroken cells and insoluble fraction were removed. Then, His$_6$-PhoP was isolated from cell lysates by passage over a Ni-NTA Starose 6 Fast Flow column (Nanotion Biotech., Suzhou, China) pre-equilibrated with binding buffer (50 mM Tris-HCl, 100 mM NaCl,

10% glycerin, 5 mM imidazole; pH 8.0), washing with the same buffer followed by the buffer with 50 mM imidazole, and then gradient elution with 50 to 500 mM imidazole with a gradient of 50 mM. The elution fraction containing His$_6$-PhoP was dialyzed in binding buffer to remove the high concentration of imidazole and concentrated using an ultrafilter (Merck KGaA, Darmstadt, Germany). SDS-PAGE and Western blotting with an anti-His-tag antibody were used to confirm the purified protein (ABclonal, Wuhan, China).

**DNA affinity purification library preparation followed by high-throughput sequencing.** Genomic DNA of *Riemerella anatipestifer* RA-YM was extracted using a bacterial DNA kit (Omega Bio-Tek, USA). Then, 10 $\mu$g of genomic DNA was sheared to a length of 300 to 500 bp using ultrasonication for 15 cycles with 3 s on and 7 s off on ice. The size of the sheared DNA fragment was checked with a 2% agarose gel, and then 2 $\mu$g of sheared genomic DNA was mixed with purified His$_6$-PhoP in 200 $\mu$L buffer A (10 mM Tris-HCl [pH 7.5], 1 mM dithiothreitol [DTT], 50 mM KCl, 5 mM MgCl$_2$, 25% glycerol, and 50 mM acetyl phosphate); reaction mixtures were incubated for 30 min at room temperature. Ten microliters of this reaction was transferred to a 1.5-mL tube with 70 $\mu$L buffer C (10 mM Tris-HCl [pH 7.5], 5 mM MgCl$_2$, 50 mM KCl, 25% glycerol, and 500 mM imidazole) and labeled as the input DNA. The 190-$\mu$L mixture was added for 30 min to 60 $\mu$L of Ni-NTA agarose resin that had been washed twice with buffer B (10 mM Tris-HCl [pH 7.5], 5 mM MgCl$_2$, 50 mM KCl, and 25% glycerol) via centrifugation at 100 $\times$ *g* for 2 min. The mixture was centrifuged at 100 $\times$ *g* for 2 min, washed with buffer A three times to remove the supernatant (unbound DNA), and incubated with 70 $\mu$L of buffer C at room temperature for 5 min to elute His$_6$-PhoP. After centrifugation at 100 $\times$ *g* for 2 min, the supernatant was labeled as the PhoP-bound DNA. The Rapid Plus DNA library preparation kit for Illumina (ABclonal, Wuhan, China) was used to prepare the PhoP-bound DNA and input DNA libraries for Illumina sequencing according to the manufacturer's instructions, with the adapter 96 kit (PCR index primers in dual DNA) for Illumina (ABclonal, Wuhan, China). Each sequencing was performed on an Illumina NovaSeq 6000 platform with up to 6 Gb data with paired-end 150-nt reads (PE150) at Annoroad Gene Technology.

**RNA extraction and cDNA preparation.** RA-YM and the Δ*phoP* and Δ*phoR* strains were cultured on TSA medium, respectively; for each strain, a single colony was cultured in TSB medium at 37°C overnight and then transferred to TSB medium at dilutions of 1:100. After shaking at 37°C and 200 rpm, the bacteria were harvested by centrifugation when the OD$_{600}$ reached 0.8. RNA was extracted from the harvested sample, using a bacterial RNA kit. cDNA was obtained by reverse transcription-PCR of RNA using HiScript II reverse transcriptase (Vazyme, Nanjing, China).

**RNA-seq.** RA-YM and the Δ*phoP* and Δ*phoR* strains were cultured on TSA medium; for each strain, a single colony was cultured in TSB medium at 37°C overnight and then transferred to TSB medium at dilutions of 1:100. After shaking at 37°C and 200 rpm, the collected bacterial RNA was extracted when the OD$_{600}$ reached 0.8. RNA samples were sent to Wuhan Bena Technology Co., Ltd., for transcriptome sequencing and analysis.

**Real-time qPCR.** Total RNA from RA-YM and its derivatives were extracted and reverse transcribed into cDNA as described above. qPCR was performed in technical duplicates with 5 $\mu$L ChamQ universal SYBR qPCR master mix (Vazyme, Nanjing, China), 0.25 $\mu$L of each primer (10 $\mu$M; listed in Table S2), and 4.5 $\mu$L diluted cDNA sample in a 96-well PCR plate (Thermo Scientific, Massachusetts, USA). The plate was run in a Bio-Rad CFX96 machine (Bio-Rad, Hercules, CA, USA). *recA* was chosen as the reference gene. Results were analyzed with the comparative critical threshold cycle method.

**Analysis of DAP-seq data and RNA-seq data.** The quality of the raw paired-end reads from DAP-seq and RNA-seq was evaluated using FASTQC (www.bioinformatics.babraham.ac.uk/projects/fastqc/), and then low-quality reads and contamination were filtered using Trimmomatic (64). The clean reads were aligned to the *Riemerella anatipestifer* RA-YM genome using Bowtie2 with default parameters. Unmapped reads and nonuniquely mapped reads (mapping quality <30) were removed and PCR duplicate reads were removed with SAMtools (65). To comprehensively analyze the multi-omics data from DAP-seq and RNA-seq, deepTools2 (66) was utilized to count the coverage through the whole genome with a bin size of 1 Mb, and the samples were normalized by reads per kilobase per million mapped reads. Statistical analysis and data records obtained with Student's *t* tests were used to compare gene expression data. For RNA-seq, the normalization of counts and detection of DEGs were performed by DESeq2 (absolute value of fold change ≥2 and $P < 0.05$) (67) on the R platform. For DAP-seq, MACS2 (68) was used to call the peaks from data with the following parameters: -f BEDPE -g 2100000 -B -q 0.01. DeepTools was used to generate a bigWig file based on the comparison of two normalized BAM files, and the continuous data were visualized in TBtools (69).

**EMSA.** The predicted motif located on the upstream region of candidate genes was synthesis as a biotin-labeled probe via annealing after 95°C for 2 min of a pair of reverse complementary primers, one of which was 5′ modified by biotin. All primers, including biotin-modified primers, were synthesized by Tsingke (Beijing, China) and are listed in Table S2. The chemiluminescent EMSA kit (Beyotime Biotechnology, Shanghai, China) was used according to the manufacturer's instructions. One hundred nanograms of biotin-labeled probe was mixed with 30 to 240 nM His$_6$-PhoP at 25°C for 15 min. Precooled 0.5$\times$ Tris-borate-EDTA (TBE) was prepared and was the main buffer for electrophoresis and transformation; this was pre-electrophoresed on a nondenatured PAGE gel. The mixture was then electrophoresed on a nondenatured PAGE gel (pre-electrophoresed for an hour at 100 V) in an ice bath at 100 V for 120 min in 0.5$\times$ TBE, and the gel was transferred to an N+ nylon membrane (GE Amersham, USA) at 380 mA for 30 min in 0.5$\times$ TBE. The UV cross-linking of the N+ nylon membrane was carried out for 30 min under a UV lamp (10 cm). The cross-linked N+ nylon membrane was incubated in blocking buffer (Beyotime Biotechnology, China) on a horizontal shaker slowly for 30 min, and then the blocking buffer was removed. Next, blocking buffer containing streptavidin-horseradish peroxidase (HRP)

conjugate (Beyotime Biotechnology, China) and diluted 1:2,000 was added for 30 min on a horizontal shaker. The N+ nylon membrane was washed for 10 min four times with washing buffer (Beyotime Biotechnology, China) and then transferred to detection equilibrium solution (Beyotime Biotechnology, China) with shaking for 10 min. Last, the bands were detected by BeyoECL Plus after dyeing (Beyotime Biotechnology, China).

**Growth experiments.** The growth curves of the $\Delta phoP$ and $\Delta phoR$ strains and RA-YM in TSB were determined. Indicated strains were grown to exponential phase ($OD_{600}$ = 0.6 to 0.8) in TSB, at which point they were harvested by centrifugation and resuspended to an $OD_{600}$ of 1 in TSB and then transferred to TSB medium at the ratio of 1:100. Two hundred microliters of diluted bacteria in TSB was transferred to a 200-well honeycomb plate (Bioscreen). The plate was incubated at 37°C in Bioscreen C MBR (Bioscreen, Finland) for 48 h, and $OD_{600}$ was measured every 30 min for the duration of growth.

**Construction of $\Delta phoP$ and $\Delta phoR$ strains.** With the RA-YM genome as the template, the left- and right-arm fragments of *phoP* and *phoR* genes were amplified by PCR. According to the sequence of the spectinomycin resistance gene (Spec) in plasmid pIC 333, primers were designed for overlap PCR to amplify the resistance gene. The Spec and left- and right-arm fragments were linked by overlap extension PCR. With pRE112 as the suicide plasmid, KpnI and SacI restriction sites were applied to construct recombinant suicide plasmids. *E. coli* $\chi$7213 competent cells were transformed with pRE-PhoP-LSR or pRE-PhoR-LSR and served as the donor strain for transconjugation. Cells of the donor strain and RA-YM were washed and resuspended in 10 mM $MgSO_4$ three times, then mixed at a ratio of 4:1 ($1 \times 10^9{:}2.5 \times 10^8$), and applied dropwise onto a sterile filter membrane disc (0.45-$\mu$m diameter) which was placed on a TSA plate previously supplemented with 50 $\mu$g/mL of diaminopimelic acid, followed by incubation at 37°C for 12 h. The bacterial cells were washed off the disc and resuspended with 10 mM $MgSO_4$ three times, and the cells were plated onto TSA containing spectinomycin followed by 48 h growth at 37°C with 5% $CO_2$. The colonies were picked, and target strains were identified by PCR with the primers listed in Table S2.

**Pathogenicity analysis of $\Delta phoP$ and $\Delta phoR$ strains in ducklings.** The $\Delta phoP$ and $\Delta phoR$ strains and RA-YM were prepared in TSB medium and centrifuged at 5,000 rpm for 3 min, respectively. The strains were resuspended in phosphate-buffered saline (PBS) and centrifuged again 3 times. The $OD_{600}$ values of the bacteria were determined. The bacterial solution was diluted to $5.0 \times 10^9$, $5.0 \times 10^8$, $5.0 \times 10^7$, $5.0 \times 10^6$, and $5.0 \times 10^5$ CFU/mL. Twelve-day-old Cherry Valley ducks were divided into 16 groups, 10 in each group. The specific grouping is shown in Table 2. Each flipper was injected with 0.2 mL bacterial solution, and the control group was injected with the same amount of PBS. The apparent changes in ducklings after bacterial injection were observed. Deaths were recorded, and $LD_{50}$ was calculated. Twelve-day-old Cherry Valley ducks were divided into three groups. The $\Delta phoP$ and $\Delta phoR$ strains and RA-YM were prepared as described above. Each duck was inoculated with 0.2 mL ($1.0 \times 10^5$ CFU) bacterial solution, and the control group was injected with the same amount of sterilized PBS. After 24 h and 48 h of inoculation, 5 ducklings were randomly selected for calculating bacterial load in tissues and blood. Refer to published literature for details of the methods (17). The ducklings infected with the $\Delta phoP$ and $\Delta phoR$ strains and RA-YM were dissected to observe the pathological changes, especially in the brain, heart, liver, and spleen. The tissues were fixed in 10% formalin for pathological section.

**Data availability.** RNA-seq raw data have been deposited in the NCBI SRA database with study accession numbers SRR14321696 to SRR14321704, and DAP-seq raw data have been deposited with numbers SRR18392563 to SRR18392566. Original DAP-seq data were uploaded to the SRA with the accession number PRJNA818095. The RA-YM genome assembled on the PacBio platform has been deposited in NCBI/GenBank under the accession number CP079205.1.

## SUPPLEMENTAL MATERIAL

Supplemental material is available online only.

**SUPPLEMENTAL FILE 1**, PDF file, 1.2 MB.

## ACKNOWLEDGMENTS

This study was supported by the National Natural Science Foundation of China No.31872498 (Z.L.). The funders had no rule in the design or execution of the experiments described.

We are grateful to all members of the Li laboratory for helpful discussion and support. We thank Ke Xiao (Huazhong Agricultural University) for technical and equipment assistance on DNA library construction and multi-omics research. We thank Jianming Zeng (University of Macau) for generously sharing their experience and codes.

Conceptualization: Yang Zhang, Ying Wang. Data curation: Yang Zhang, Ying Wang. Formal analysis: Yang Zhang, Ying Wang, Zili Li. Funding acquisition: Zili Li. Investigation: Yang Zhang, Ying Wang, Yanhao Zhang, Xiangchao Jia, Chenxi Li. Methodology: Yang Zhang, Ying Wang, Zutao Zhou, Sishun Hu, Zili Li. Project administration: Yang Zhang, Ying Wang, Sishun Hu, Zili Li. Resources: Yang Zhang, Ying Wang, Zutao Zhou, Zili Li. Software: Yang Zhang. Supervision: Yang Zhang, Zili Li. Validation: Yang Zhang, Ying Wang. Visualization: Yang Zhang, Ying Wang. Writing – original draft: Yang Zhang, Ying Wang. Writing – review & editing: Yang Zhang, Zili Li.

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
