## [Reviewer comments · Microbiology Spectrum]

Microbiology Spectrum

Genome-wide analysis reveals PhoP regulates pathogenicity in *Riemerella anatipestifer*

Yang Zhang, Ying Wang, Yanhao Zhang, Xiangchao Jia, Chenxi Li, Zutao Zhou, Sishun Hu, and Zili Li

Corresponding Author(s): Zili Li, Huazhong Agricultural University

Review Timeline:

Submission Date:	May 31, 2022
Editorial Decision:	July 19, 2022
Revision Received:	September 14, 2022
Accepted:	September 15, 2022

Editor: Artem Rogovskyy

Reviewer(s): The reviewers have opted to remain anonymous.

Transaction Report:

DOI: <https://doi.org/10.1128/spectrum.01883-22>

July 19, 2022

Prof. Zili Li
Huazhong Agricultural University
College of Veterinary Medicine
No.1, Shizishan Street
Wuhan, Hubei 430070
China

Re: Spectrum01883-22 (Genome-wide analysis reveals PhoP regulates pathogenicity in *Riemerella anatipestifer*)

Dear Prof. Zili Li:

Link Not Available

Sincerely,

Artem Rogovskyy

Journals Department
Reviewer comments:

Reviewer #1 (Comments for the Author):

1. Introduction. I think this has too much detail and is more like a literature review. Much of the detail could be removed without compromising clarity.
2. Line 84. What is a "reaction protein"? Is meant response regulator?
3. Line 85. "causes the expression" is poor English. What is meant is regulates the expression...? And this can be positive or negative.
4. Line 86. Better English: discovering the genes regulated by....
5. Line 90. It would be useful to discuss how these are different from PhoP/PhoR. It will help bring out the significance of the

subsequent statement that PhoP/PhoR in *R. anatipestiferis* first described for Gram negative.

6. Line 94. This is confusing to include the genomic loci. They are not defined as such. Perhaps just the gene names and the reference to the prior study make this more clear for the reader?

7. Line 98? What is meant by speculated? The data cited establishes a role in pathogenesis.

8. Line 99. It would help the reader to include more details as to just how it was concluded that these loci are functionally PhoP/PhoR? And how this is differentiated from PhoP/PhoQ or the aerotolerance TCS systems that comparisons will be made to.

9. Line 104. What is RR? Define abbreviations.

10. Line 107. Drawbacks is meant, not defects?

11. Line 108. What is a ChIP-level antibody? It will help the reader to avoid jargon.

12. Line 114. This would be a good place to define the abbreviation DAP-seq.

13. Line 121. Provides a foundation is better English than laid.

14. Line 121 Understanding is better English than clarifying

15. Line 123 Insights is better English than sight

16. Line 124. Provides (present tense) vs provided (past tense) is better English. Conclusions are most appropriately made in present tense, while reference to data are most appropriate in past tense.

17. Line 125. See line 124

18. Line 127. No context is provide to make sense of this conclusion that these data are important for understanding cross-talk in other pathogens? More detail would help to provide context.

19. Line 140. What is meant by "self-made"?

20. Line 143. This is expansive. The presentation of negative data here does not add to the presentation. There could be many reasons for the failure to produce a suitable antibody and none are important for understanding the DAP-seq experiment.

21. Line 143-144. A stronger presentation is not to focus on the negatives and the failure to produce an antibody, but instead, to focus on the advantages of the DAP-seq approach.

22. Line 147. Was this protein phosphorylated or not? Is not clear.

23. Line 147. some details on the DAP-seq experiment are also required here, including he specific strain and negative controls to establish specificity of the His6-PhoP to DNA.

24. Line 147. The quality of these results using this method is completely dependent on the quality of the purified protein. Thus, it will be essential to include details as to the quality of this protein preparation, including it purity, its ability to bind specifically to DNA as compared to an appropriate negative control, that it is of the appropriate size and that it actually is the PhoR protein and not some fortuitously purified protein.

25. Similar to above. Characterization of the truncated DBD protein is also important for evaluation of these results.

26. Line 147. Details are required in order to evaluate the quality of these data. Including some specific details of the method, the specific strain that was analyzed, the stage of growth that were sampled and the number of technical and biological replicates

27. Line 148. What is meant by random? At best you can conclude that no pattern or order was apparent in the distribution?

28. Line 149. Reference for the MEME tools should be included.

29. Line 163. How is it concluded that all binding to the 764 genes was specific? What negative controls establish the specificity of binding?

30. Line 167. It is not possible to conclude from these data that the domain is not involved. Only that it is not sufficient for binding. It still could be necessary.

31. Line 176. These data are not sufficient to support a conclusion that phosphorylation is not important for binding DNA. The paradigm is that phosphorylation increases binding affinity, which in an in vitro assay would be sensitive to the concentration of the respective proteins. And also their stability in vitro. Thus, comparison of stability and calculation of Kd is required to support this conclusion.

32. Line 194. What is a "confident correlation"?

33. Line 197. This statement is not clear. What is meant by "changed in the same pattern"? Was this different media or stress imposed on the cultures?

34. Line 199. What is meant by "inconsistent regulation"?

35. Line 203. The rationale for this section requires more detail. Is this concordance between the two datasets? If so, whatLine 203 is the rationale for this vs possible technical differences in sensitivity that could account for non-concordance?

36. Line 203. It also appears that many more genes were found to be regulated by DAP-seq than by RNA-seq? What could explain the discrepancy? It is likely that there are many explanations that involve the strengths and weaknesses of the methods?

37. Line 203. Is the approach here that valid regulated genes are only those that appear in both data sets? That seems limited and requires more justification, as there could be many technical reasons for this. And it overlooks the result above finding the common motif in the 764 genes identified by DAP-Seq. Do the RNA-seq dataset genes share this motif?

38. Line 214. What is "membrane compromised TBDR"?

39. Line 219. This section is more a discussion rather than a presentation of data.

40. Line 234. Do the mutants have a defect in aerotolerance? That seems like it would be easy to test and these data would significantly test this hypothesis.

-

41. Line 264. What is a gradient dose? Is meant "a range of doses"?

42. Line 256. This conclusion of "mainly regulate..." is based on the assumptions that the only valid regulation are those in the concordance of the DAP-seq and RNA-seq datasets. The strength of this conclusion remains to be established by more information, as indicated above.

43. Line 268. What is "retarded behavior"?
44. Line 269. What is "favorable mental conditions"?
45. Line 270. huddling is a mental state? Generally, this sort of behavior is usually described as moribund or lethargic, as it does not imply that there is any direct effect on brain function, only it can be concluded that the animals are experiencing an overall state of malaise and lethargy because they are sick.
46. Line 299. "Mock infected" is more precise than "infected by PBS"
47. Line 350. As discussed above, there is no increase in clarity from the inclusion of negative data. There are many reasons why an attempt to make a highly specific antibody was not successful. None of which are important for interpretation of the actual data presented.
48. Line 395. As discussed above, additional data are required to support this conclusion.
49. Line 396. Assuming the conclusion is correct, that phosphorylation has no effect, then what is a phosphorylation-independent model that can explain regulation of these genes?

Reviewer #2 (Comments for the Author):

Zhang et al. have examined the regulon of a two-component system in *Riemerella anatipestifer*. They already studied this TCS, that they then called PhoP-PhoR, in a previous study and found that it was important for pathogenicity of this duck pathogen. In this work, they go in depth in the determination of the regulon by using RNA-seq and DAP-seq to delineate the full direct regulon and confirm relevance to virulence. While the technical aspects seem sound and some of the high-throughput findings are confirmed with targeted approaches, I have few concerns that would need to be addressed for this study to be considered for publication.

Major comments :

- The authors have already published a study on the regulon of this two-component system (<https://doi.org/10.3389/fmicb.2017.00688>). In the present manuscript, they re-perform transcriptomics and perform additional DAP-seq and confirmation experiments. Both studies also present virulence data on mutants of this system. They should thus make clearer what was the rationale for the need of these additional approaches and what does this new study actually bring in terms of new knowledge and biological understanding. Several of the main conclusions (i.e. the PhoPR regulon has been identified & the PhoPR system is essential for virulence in duck) are redundant between the two papers, what is actually new should be highlighted better if the authors want to claim this study as bringing novel insights.
- The authors describe this TCS as a PhoPR system that senses environmental phosphate. This claim is based solely on computational prediction and no experimental proof was made in the current or previous study. Now that they have mutants for both the response regulator and histidine kinase, and primers for RT-qPCR of their targets, it should be relatively easy to test that claim in low or high phosphate growth conditions. If it is found that this TCS does not respond to phosphate availability, renaming or at least discussion of these findings should be considered.
- The authors naming of this TCS as a PhoPR system was notably based on a KEGG / GO pathway enrichment approach from their previous RNA-seq results. In the current, much more well defined version of the regulon, what would be the result of such analysis ?
- The authors state that in their previous study they found the PhoPR system to regulate one third of genes using RNA-seq on mutants. In the present study and using a similar approach, they find a much lower number of dysregulated genes (even when combining results from both mutants), even though they use a more complete version of the genome. This difference should be discussed.
- The main method used in this study is DAP-seq, however the authors do not show anywhere how the data actually looks like. I would suggest adding a representation of the actual peaks, at least for some of the examples studied in depth (those with EMSA for instance) in order for the reader to appreciate quality and intensity of the signal, as is usually done in studies reporting DAP- or ChIP-seq data.
- The authors have resequenced their bacterium to provide a more complete assembly of the genome. As this new assembly is used as a reason for the need of reanalyzing the TCS regulon, the differences brought by this new assembly should be briefly discussed.

Minor comments :

- Concerning the virulence phenotype, can the authors comment on what is the reason for the decreased virulence in the PhoPR mutants ? i.e. which target genes could be responsible ?

- Statistical analysis should be performed in figure 8
- DAP-seq is an emerging method for the study of TF binding sites that is not yet well known, especially in studies on prokaryotes, attention should thus be paid to properly cite the method and previous works. Particularly, the authors do not cite papers from the O'Malley group, which developed DAP-seq, but only older versions of the DAP-chip approach. At least the Nature protocol DAP-seq paper should be cited (<https://doi.org/10.1038/nprot.2017.055>). Additionally, the authors use a version of the protocol that is made for response regulators and uses acetyl phosphate, which has already been used in other bacterial studies (ex: <https://doi.org/10.1093/nar/gkab928> ; <https://doi.org/10.1111/mmi.13909>) which should also be considered as references.
- Line 462: "The present study has reported the first application of the combination of DAP-seq with RNA-seq ...": that claim is not true (see <https://doi.org/10.1111/mmi.13909> for an example of bacterial TCSs, and <https://doi.org/10.1128/mSystems.00753-20> for bacterial OCSs). Citing previous similar studies should thus be considered.
- Figure 5 could be made a supplementary figure.
- Line 61 : Should remove dot between end of sentence and reference
- Should homogenize way of referring to papers in the text, ex: "Dou reported..." or "Tian et al. identified...". I suggest using et al. in all instances
- Line 400 & 402: "phoP and phoP"
- Generally, the manuscript should be thoroughly proofread again for typos and confusing phrasings

Staff Comments:

Preparing Revision Guidelines

Please return the manuscript within 60 days; if you cannot complete the modification within this time period, please contact me. If you do not wish to modify the manuscript and prefer to submit it to another journal, please notify me of your decision immediately so that the manuscript may be formally withdrawn from consideration by Microbiology Spectrum.

Dear editors and reviewers:

Thank you for your decision and constructive comments on my manuscript.

We have carefully considered the suggestion of Reviewer and make some changes.

We have tried our best to improve and made some changes in the manuscript. We

submitted the revised version with '**Marked Up Manuscript – For Review Only**'

that indicates the changes from the original submission and a clean and new

'**Manuscript**' without changes marked.

We marked the modified statement in red and provide two types of Line number according to the comments, the yellow Line number in brackets is for

Manuscript. Revision notes, point-to-point, are given as follows:

To Reviewer #1:

1. Introduction. I think this has too much detail and is more like a literature review.

Much of the detail could be removed without compromising clarity.

Answer: Thank you for the suggestion. We have removed some content about the researches about *R. anatipestifer*, and modified some contents according to the comment.

2. Line 84. What is a "reaction protein"? Is meant response regulator?

Answer: Thank you for the comment. Yes, we meant 'response regulator' . We have modified it as '**response regulator**' throughout the text.

3. Line 85. "causes the expression" is poor English. What is meant is regulates the expression...? And this can be positive or negative.

Answer: We apologize for the poor language of our manuscript. We really appreciate the suggestion. We have modified the sentence as 'The phosphorylated response regulator upregulates or downregulates the expression of the bacterial genes' in Line 84-86 (72-73).

4. Line 86. Better English: discovering the genes regulated by...

Answer: Thank you for the suggestion. We have modified the sentence as 'Discovering the gene regulated by TCSs of bacterial pathogens is essential for understanding the mechanisms of bacterial survival and infection' in Line 87-88(74-75).

5. Line 90. It would be useful to discuss how these are different from PhoP/PhoR. It will help bring out the significance of the subsequent statement that PhoP/PhoR in *R. anatipestiferis* first described for Gram negative.

Answer: Thank you for the suggestion. We have rewritten this part as 'PhoPR and PhoPQ mainly exist in Gram-positive bacteria and Gram-negative bacteria respectively, and PhoR and PhoQ showed a high similarity in their C-terminal portion. Although many research reported PhoPR and PhoPQ are required for the pathogenicity in their respective species, they are not the physiological equivalent.

PhoPR in Gram-positive bacteria is mainly activated in low phosphate conditions, regulating gene expression to cope with low phosphate environments, while the PhoPR in *M. tuberculosis* does not respond to the phosphate. In Gram-negative bacteria, PhoPQ mainly responds to the magnesium limitation and antibacterial peptides, and phosphate starvation is sensed by PhoBR.' in Line 88-97(75-84).

6. Line 94. This is confusing to include the genomic loci. They are not defined as such. Perhaps just the gene names and the reference to the prior study make this more clear for the reader?

Answer: Thank you for the suggestion, we modified as '*phoPR* double gene deletion strain of *R. anatipestifer* was constructed in our previous research' in Line 104-105 (84-85).

7. Line 98? What is meant by speculated? The data cited establishes a role in pathogenesis.

Answer: Thank you for the suggestion. It's inappropriate description here, and we have modified and added the data of virulence reduction by deletion of PhoPR as '*the double gene deletion strain completely lost its pathogenicity to ducklings (LD50 > 10¹¹ CFU)*' in Line 115(91-92).

8. Line 99. It would help the reader to include more details as to just how it was concluded that these loci are functionally PhoP/PhoR? And how this is

differentiated from PhoP/PhoQ or the aerotolerance TCS systems that comparisons will be made to.

Answer: Thank you for the suggestion. This conclusion is from the previous study that PhoP/PhoR TCS was predicted by KEGG pathway and we added the detailed and reference. We added the detail of gene expression in high (1 mM) or low (1 μ M) phosphate, and displayed the root neighbor-joining tree of the amino acid sequences of PhoP and PhoR respectively. We have modified the sentence as 'Analysis of RNA-seq using KEGG pathways and the upregulation of PhoPR in phosphate starvation indicated that RAYM_RS09735/RAYM_RS09740 was the PhoP/PhoR two-component system' in Line 109-111 (86-88).

RT-qPCR analysis of *phoR* and *phoP* in low (1 μ M) and high (1 mM) phosphate growth condition.

The root neighbor-joining tree of PhoP/PhoR, PhoB/PhoR, PhoP/PhoQ and PhoP/PhoR of *R.*

anatipestifer.

(A) The analysis of the evolutionary relationship between PhoP of RA, PhoB of Gram-negative bacteria and PhoP. (B) The analysis of the evolutionary relationship between PhoR of RA, PhoR of gram-positive bacteria and PhoQ of Gram-negative bacteria.

9. Line 104. What is RR? Define abbreviations.

Answer: Thank you for the suggestion. RR is response regulator. We added the abbreviation in Line 84(71).

10. Line 107. Drawbacks is meant, not defects?

Answer: Thank you for the suggestion. We have deleted this part of discussing about the drawbacks of ChIP-seq according to the comment.

11. Line 108. What is a ChIP-level antibody? It will help the reader to avoid jargon.

Answer: Thank you for the suggestion. We have removed this part in Introduction according to Comment 1.

12. Line 114. This would be a good place to define the abbreviation DAP-seq.

Answer: Thank you for the suggestion, we added the abbreviation in Line 128(98) according to the comment.

13. Line 121. Provides a foundation is better English than laid.

Answer: Thank you for the suggestion, and we have modified the sentence as

'Our data provides a foundation' in Line 135(105).

14. Line 121 Understanding is better English than clarifying

Answer: Sincerely thank you for this suggestion. We modified the sentence as

'Our data provides a foundation for understanding the role of the *phoP/phoR* two-component system in the pathogenic process of *R. anatipestifer*' in Line 135-137(105-106) according to this comment.

15. Line 123 Insights is better English than sight

Answer: Thank you for the suggestion. We modified the sentence as 'a new insight into the regulation of PhoP' in Line 137(107).

16. Line 124. Provides (present tense) vs provided (past tense) is better English.

Conclusions are most appropriately made in present tense, while reference to data are most appropriate in past tense.

Answer: Thank you for the suggestion, and we modified as 'Our data provides a foundation for understanding the role of the *phoP/phoR* two-component system in the pathogenic process of *R. anatipestifer*, a new insight into the regulation of PhoP to aerotolerance in *R. anatipestifer*, and a theoretical basis for discovering new drug targets.' in Line 135-139(105-108) according to the comment.

17. Line 125. See line 124

Answer: Thank you for the suggestion. We have modified it as 'This direct and global exploration of the regulation of *phoP* provides a model for gene regulation in *R. anatipestifer* and other pathogens' in Line 139-141(108-110).

18. Line 127. No context is provide to make sense of this conclusion that these data are important for understanding cross-talk in other pathogens? More detail would help to provide context.

Answer: Thank you for the suggestion. Sorry for the inappropriate sentence, and we have removed the sentence 'and might explain how these response regulators cross-talk in other pathogens' .

19. Line 140. What is meant by "self-made"?

Answer: Thanks for the suggestion. We meant that we emulsified PhoP and injected into rabbits, and purified the anti-PhoP antibody from the serum. And we removed this part according to the comment 20, 21.

20. Line 143. This is expansive. The presentation of negative data here does not add to the presentation. There could be many reasons for the failure to produce a suitable antibody and none are important for understanding the DAP-seq experiment.

Answer: Thank you for the constructive suggestion. We have removed the negative data, and rewrite as 'The DNA-affinity-purified sequencing (DAP-seq)

was utilized for investigating the PhoP-binding region on the whole genome of *R. anatipestifer* which could effectively enriched the binding peaks of the phosphorylated PhoP, and avoid the disadvantage of lacking specific antibodies and indirect PhoP-binding in vivo.' in Line 154-157(122-125).

21. Line 143-144. A stronger presentation is not to focus on the negatives and the failure to produce an antibody, but instead, to focus on the advantages of the DAP-seq approach.

Answer: Thank you for the constructive suggestion. We have removed the sentences, and rewrite the advantages of DAP-seq as 'DAP-seq allows us to use phosphoryl group donor to mimic phosphorylation of RR, avoid the limitation of unknow activating signals of PhoP. DAP-seq does not require a specific antibody or tagged transgenic lines comparing with ChIP-seq, and only requires an affinity method for the tag-labeled response regulator.' in Line 405-409(323-327).

22. Line 147. Was this protein phosphorylated or not? Is not clear.

Answer: Yes, we have added the details in the text as 'DAP-seq was performed with phosphorylated recombinant His₆-PhoP' in Line 175-176(135-136).

23. Line 147. some details on the DAP-seq experiment are also required here, including he specific strain and negative controls to establish specificity of the His6-PhoP to DNA.

Answer: Thank you for the suggestion. We have added the information required and modified as 'The recombinant His₆-PhoP was expressed by the *E. coli* BL21 transferred with pET-28a-PhoP, and purified via Ni-NTA resin (Fig. S1)' and 'compared with the negative control performed without His₆-PhoP' in Line 174(134) and 180(140).

24. Line 147. The quality of these results using this method is completely dependent on the quality of the purified protein. Thus, it will be essential to include details as to the quality of this protein preparation, including its purity, its ability to bind specifically to DNA as compared to an appropriate negative control, that it is of the appropriate size and that it actually is the PhoR protein and not some fortuitously purified protein.

Answer: Thank you for the suggestion. We added the information about the purity of PhoP in supplement. In Figure S1 we provided the SDS-PAGE result and added chromatography of purified PhoP to show the purity and size, and we mentioned the detail of purification in Materials and methods. Negative control of DAP-seq we performed is that the sheared genome directly binding with resin, and the eluted DNA is the non-PhoP binding. We added the information of purity in Fig. S1, 'Chromatography of His₆-PhoP by gel-filtration. The number above the peaks is the retention time. The purity of His₆-PhoP is 94.4232%' .

25. Similar to above. Characterization of the truncated DBD protein is also important for evaluation of these results.

Answer: Thank you for the suggestion. We have updated the information of the purity of PhoP-DBD in supplement according to the comment. We added the information of purity in Fig. S1, 'Chromatography of His₆-PhoP-DBD by gel-filtration. The number above the peaks is the retention time. The purity of His₆-PhoP-DBD is 91.4392%.'

26. Line 147. Details are required in order to evaluate the quality of these data.

Including some specific details of the method, the specific strain that was analyzed,

the stage of growth that were sampled and the number of technical and biological replicates

Answer: Thank you for the suggestion. We added the details required in in Line 175-180(135-140), 'DAP-seq was performed with phosphorylated recombinant His₆-PhoP incubated with the sheared genome of RA-YM in two duplicates, and the high-throughput data were generated from the PhoP-binding DNA. The DAP-seq analysis of phosphorylated His₆-PhoP identified 583 enriched peaks covering the upstream regions of 764 genes, compared with the negative control performed without His₆-PhoP' , and mentioned the detailed protocol in Material and Methods.

27. Line 148. What is meant by random? At best you can conclude that no pattern or order was apparent in the distribution?

Answer: Thank you for underlining this deficiency. We modified the sentence as 'These peaks were distributed along the *R. anatipestifer* genome with no apparent pattern as shown in Figure 1' in Line 180-182(140).

28. Line 149. Reference for the MEME tools should be included.

Answer: Thank you for the suggestion. We added the reference in Line 182(143).

29. Line 163. How is it concluded that all binding to the 764 genes was specific?

What negative controls establish the specificity of binding?

Answer: Thank you for the suggestion. Several statements that we made were more ambiguous than intended. We added the statements that the negative control of DAP-seq was performed without PhoP, which means, the high-throughput data of negative control was generated from the unspecific bound DNA eluted from the resin. The 764 genes were enriched based on the nonspecific binding. And we validated the results by perform EMSA with the upstream region of 8 genes.

30. Line 167. It is not possible to conclude from these data that the domain is not involved. Only that it is not sufficient for binding. It still could be necessary.

Answer: Thank you for the constructive suggestion. We modified the sentence as 'only the DNA-binding domain (DBD) of PhoP is not sufficient for the DNA binding function' in Line 201-202(161-162) according to the comment.

31. Line 176. These data are not sufficient to support a conclusion that phosphorylation is not important for binding DNA. The paradigm is that phosphorylation increases binding affinity, which in an in vitro assay would be sensitive to the concentration of the respective proteins. And also their stability in vitro. Thus, comparison of stability and calculation of Kd is required to support this conclusion.

Answer: Thank you for the suggestion. We modified the sentence as 'The results indicated that the unphosphorylated PhoP of *R. anatipestifer* could bind DNA in

in vitro in Line 215-216(174-175), and added EMSA in Figure 4 according to the comment. We added the calculation of K_d , ' K_d was calculated for Figure 4C and 4D, and the dissociation constant is the protein concentration that resulted in 50% DNA bound with PhoP. PhoP bound the upstream region of KYF39_00915 with the K_d value 29.81 nM, while the phosphorylated PhoP with the K_d value 33.87 nM. The K_d value is 39.19 nM and 39.55 nM (phosphorylated) in binding with the upstream region of KYF39_00905' in Line 210-215(169-174).

32. Line 194. What is a "confident correlation"?

Answer: Thank you for the suggestion. We have modified this sentence as

'confirmed the accuracy of the high-throughput results' in Line 235-236(194),

and added the correlation between the RT-qPCR and RNA-seq results of 8 candidate genes in Fig. S4.

33. Line 197. This statement is not clear. What is meant by "changed in the same pattern"? Was this different media or stress imposed on the cultures?

Answer: Thank you for pointing out this indistinctness. We mean that the 35 genes downregulated in both $\Delta phoP$ and $\Delta phoR$, and the 22 genes upregulated in both $\Delta phoP$ and $\Delta phoR$. We modified the sentence as 'Further analysis of these 59 genes showed that the expression of 57 genes changed in the same trend including 35 genes downregulated and 22 genes upregulated in both $\Delta phoP$ and $\Delta phoR$ ' in Line 239-241(196-199).

34. Line 199. What is meant by "inconsistent regulation"?

Answer: Thank you for pointing out this indistinctness. We meant that the 2 genes displayed different trend of regulation in $\Delta phoP$ and $\Delta phoR$, and we have

modified the sentence as 'only two genes displayed different trend of regulation' in Line 241-242(199-200).

35. Line 203. The rationale for this section requires more detail. Is this concordance between the two datasets? If so, whatLine 203 is the rationale for this vs possible technical differences in sensitivity that could account for non-concordance?

Answer: Thank you for the suggestion. We added the information to make the section clear. 'From the RNA-seq of *ΔphoP*, we can know the genes directly or indirectly regulated by PhoP, and the DAP-seq of PhoP provides information of PhoP-binding sites on the genome' in Line 249-251(205-208). We generated 183 differentially expressed genes ($|\log_2\text{FoldChange}| > 1$, $P_{\text{adj}} < 0.05$) from the RNA-seq of *ΔphoP*, and 764 genes from DAP-seq. We focused on the DEGs which the PhoP binds in the upstream region, and validated 5 genes via EMSA and RT-qPCR. We think the difference of sequencing depth and intrinsic decreased accuracy of high-throughput sequencing leads to the inconsistency between of the peak-enrichment and gene expression.

36. Line 203. It also appears that many more genes were found to be regulated by DAP-seq than by RNA-seq? What could explain the discrepancy? It is likely that there are many explanations that involve the strengths and weaknesses of the methods?

Answer: Thank you for the comment. Due to the principle of DAP-seq, the data generated from DAP-seq is the sites of PhoP-binding motif on the genome. RNA-seq mainly showed transcription level of all genes in RA-YM, and there are direct and indirect regulation caused by gene disruption. Strict statistical screening of DEGs ($|\log_2\text{FoldChange}| > 1$, $\text{P}_{\text{adj}} < 0.05$) also lead to some regulated genes not participating in the following analysis.

37. Line 203. Is the approach here that valid regulated genes are only those that appear in both data sets? That seems limited and requires more justification, as there could be many technical reasons for this. And it overlooks the result above finding the common motif in the 764 genes identified by DAP-Seq. Do the RNA-seq dataset genes share this motif?

Answer: Yes, the candidate target genes are those appear in both data sets. And we admit that there are some limitations, and we can't announce that we have screened all the direct regulatory genes. We performed EMSA and RT-qPCR on selected 5 of candidate target genes to verify the confidence of data, and showed the binding region of each gene. MEME analysis of all 583 peaks (located at the upstream of 764 target genes), showed the conserved motif appeared in 579 peaks, and absent in 4 peaks (located at upstream of 4 target genes). Not all the RNA-seq dataset genes were shared the motif, but the 50 candidate target genes we screened share the binding motif.

38. Line 214. What is "membrane compromised TBDR"?

Answer: Thank you for the suggestion, here is our writing mistake. We have modified it as 'membrane composed of TBDR' in Line 497(392).

39. Line 219. This section is more a discussion rather than a presentation of data.

Answer: Thank you for the suggestion. We moved some content to the Discussion.

40. Line 234. Do the mutants have a defect in aerotolerance? That seems like it would be easy to test and these data would significantly test this hypothesis.

Answer: Thank you for the constructive suggestion. We did perform a preliminary study on the difference of aerotolerance of mutants. We tested the change of expression level of Bat operon after H₂O₂ treatment. We added the survival rate in Discussion and Figure S5, 'the survival rate of *ΔphoP* and *ΔphoR* decreased compared with WT after exposure of 10mM H₂O₂'. We have added the content as 'The genes *batA* and *batC* were upregulated when H₂O₂ was added, and downregulated either when treated anaerobic, or when *phoP* or *phoR* was deleted, the survival rate of *ΔphoP* and *ΔphoR* decreased compared with WT after exposure of 10 mM H₂O₂' in Line 259-264(218-221). We also are conducting future work on Bat operon and constructing gene knockout strains in our future project.

41. Line 264. What is a gradient dose? Is meant "a range of doses"?

Answer: Thank you for the suggestion. We infected with RA at doses of 1×10^5 , 1×10^6 , 1×10^7 , 1×10^8 , 1×10^9 CFU per duckling, and modified it as to '7-day-old ducklings were infected with a range of doses (1×10^5 - 10^9 CFU per duckling) of ΔphoR and ΔphoP strains via fin injection' in Line 322-323 (242-243).

42. Line 256. This conclusion of "mainly regulate..." is based on the assumptions that the only valid regulation are those in the concordance of the DAP-seq and RNA-seq datasets. The strength of this conclusion remains to be established by more information, as indicated above.

Answer: Thank you for the significant suggestion. Due to the difference of sequencing depth and principle between DAP-seq and RNA-seq, there is no significant concordance in the enrichment of peaks from DAP-seq and the expression change from RNA-seq. We verified seven genes by EMSA and RT-qPCR. We removed the conclusion according to the comment.

43. Line 268. What is "retarded behavior"?

Answer: We are sorry for the confused description. We have modified the sentence as 'The surviving ducklings returned to a normal diet after 7 days of bradykinesia' in Line 326-327(246).

44. Line 269. What is "favorable mental conditions"?

Answer: We are sorry for the confused description, and we have modified the sentence as 'The ducklings infected with the $\Delta phoR$ strain displayed good mental conditions' in Line 327-328(247).

45. Line 270. huddling is a mental state? Generally, this sort of behavior is usually described as moribund or lethargic, as it does not imply that there is any direct effect on brain function, only it can be concluded that the animals are experiencing an overall state of malaise and lethargy because they are sick.

Answer: Thank you for the suggestion. We have modified the description of symptoms as 'The ducklings infected with the $\Delta phoP$ strain showed lethargic after injection, preferred huddling together than moving' in Line 329-331(248-249).

46. Line 299. "Mock infected" is more precise than "infected by PBS"

Answer: Thank you for the suggestion. We have modified the sentence as 'The

tissues did not show obvious pathological changes in mock infected, $\Delta phoP$ and $\Delta phoR$ in Line 359(277-278).

47. Line 350. As discussed above, there is no increase in clarity from the inclusion of negative data. There are many reasons why an attempt to make a highly specific antibody was not successful. None of which are important for interpretation of the actual data presented.

Answer: Thank you for the constructive comment. We have removed the part of negative data, and rewritten this part as ‘DAP-seq allows us to use phosphoryl group donor to mimic phosphorylation of RR, avoid the limitation of unknown activating signals of PhoP. DAP-seq does not require a specific antibody or tagged transgenic lines comparing with CHIP-seq, and only requires an affinity method for the tag-labeled response regulator.’ in Line 405-409(323-327) according to the comment.

48. Line 395. As discussed above, additional data are required to support this conclusion.

Answer: Thank you for the suggestion, and we have removed the wrong statement and modified the sentence as ‘Our analysis of PhoP-binding suggested that PhoP could bind DNA without phosphorylation in vitro.’ in Line 465-467(361-362) and added the K_d of DNA-binding.

49. Line 396. Assuming the conclusion is correct, that phosphorylation has no affect, then what is a phosphorylation-independent model that can explain regulation of these genes?

Answer: Thank you for the suggestion. We speculated that PhoP might have a certain binding ability to target DNA, and phosphorylation affected the binding ability in vivo, resulting in the regulation of downstream genes. This is also mentioned in the research of *M. tuberculosis* (<https://doi.org/10.1021/bi2005575>) and *B. subtilis* (<https://doi.org/10.1128/JB.186.4.1182-1190.2004>). We believed that future studies can be directed at whether phosphorylation affects the binding motif preference of PhoP, or whether phosphorylation has a greater effect on the PhoP in vivo.

To Reviewer #2:

Major comments :

- The authors have already published a study on the regulon of this two-component system (<https://doi.org/10.3389/fmicb.2017.00688>). In the present manuscript, they re-perform transcriptomics and perform additional DAP-seq and confirmation experiments. Both studies also present virulence data on mutants of this system. They should thus make clearer what was the rationale for the need of these additional approaches and what does this new study actually bring in terms

of new knowledge and biological understanding. Several of the main conclusions (i.e. the PhoPR regulon has been identified & the PhoPR system is essential for virulence in duck) are redundant between the two papers, what is actually new should be highlighted better if the authors want to claim this study as bringing novel insights.

Answer: Thank you for the constructive suggestions. As you said, in our previous research, we initially identified the *phoPR* two-component system, and performed RNA-seq and animal models tests of $\Delta phoPR$. We confirmed that *phoPR* had a great influence on the pathogenicity of RA-YM. The immunogenicity of $\Delta phoPR$ to ducklings was tested (not shown), and it was found that the pathogenicity of $\Delta phoPR$ decreased after its deletion, and its immunogenicity also decreased significantly. And we wanted to further dig the specific model of *phoPR* regulation on virulence and found an appropriate live vaccine candidate strain that retained the immunogenicity, so our work focuses on the direct regulatory pathway of PhoP in RA-YM this time. We updated the complete genome of RA-YM via Pacbio platform, and constructed *phoP* and *phoR* single-gene deletion strain, so as to more specifically study the effects of each component in the *phoPR*. We found that both $\Delta phoP$ and $\Delta phoR$ can retain immunogenicity (another article being submitted). We also obtained a further understanding of the regulation mechanism of RA-YM through the data of DAP-seq and RNA-seq. In this research, we found that the motif of PhoP-binding in RA-YM, and is mainly involved in the direct regulation of which genes, providing a foundation for the further study on

the virulence of RA.

- The authors describe this TCS as a PhoPR system that senses environmental phosphate. This claim is based solely on computational prediction and no experimental proof was made in the current or previous study. Now that they have mutants for both the response regulator and histidine kinase, and primers for RT-qPCR of their targets, it should be relatively easy to test that claim in low or high phosphate growth conditions. If it is found that this TCS does not respond to phosphate availability, renaming or at least discussion of these findings should be considered.

Answer: Thank you for the suggestion. We tested the expression of *phoP* and *phoR* in low or high phosphate growth conditions. Using RT-qPCR, we found that *phoP* and *phoR* was upregulated in low Pi medium (1 μ M), while downregulated in high Pi medium (1mM).

RT-qPCR analysis of *phoR* and *phoP* in low (1 μ M) and high (1 mM) phosphate growth condition.

- The authors naming of this TCS as a PhoPR system was notably based on a KEGG / GO pathway enrichment approach from their previous RNA-seq results. In the current, much more well defined version of the regulon, what would be the result

of such analysis?

Answer: Thank you for the suggestion. We annotated the new version genome by KEGG and GO. From the predicted results, the TCS is still annotated as PhoP and PhoR in KEGG. We tested gene expression in high- or low-phosphate, and displayed the root neighbor-joining tree of PhoP and PhoR respectively.

The root neighbor-joining tree of PhoP/PhoR, PhoB/PhoR, PhoP/PhoQ and PhoP/PhoR of *R.*

anatipestifer.

(A) The analysis of the evolutionary relationship between PhoP of RA, PhoB of Gram-negative bacteria and PhoP. (B) The analysis of the evolutionary relationship between PhoR of RA, PhoR of gram-positive bacteria and PhoQ of Gram-negative bacteria.

- The authors state that in their previous study they found the PhoPR system to regulate one third of genes using RNA-seq on mutants. In the present study and using a similar approach, they find a much lower number of dysregulated genes (even when combining results from both mutants), even though they use a more complete version of the genome. This difference should be discussed.

Answer: Thank you for the suggestion, we added the discussion according to the comment. 'From the perspective of gene or operon function, there should exist divergence in the DEGs produced by the deletion of *phoP*, *phoR* or *phoPR* due to the different regulatory pathway in cell. As a HK, PhoR not only phosphorylated the cognate RR, but also possible transferred the phosphoryl group to other RRs, while PhoP as the RR, which may also get phosphorylated by other non-cognate HKs. Therefore, we believe that there is difference in the global transcriptome between the double gene deletion strain and single-component deletion strains. Due to the limitation in previous research, the RNA-seq of $\Delta phoPR$ was not duplicated and the reference genome was not appropriate, resulting in an inability to conduct statistical analysis.' In Line 482-491 (376-385). We updated a complete genome of RA-YM by PacBio platform, and conducted three biological duplicates in RNA-seq in this study. Because of the statistical analysis of RNA-seq this time, the DEGs we obtained had a higher credibility, but reduced in number compared with previous data.

- The main method used in this study is DAP-seq, however the authors do not show anywhere how the data actually looks like. I would suggest adding a representation of the actual peaks, at least for some of the examples studied in depth (those with EMSA for instance) in order for the reader to appreciate quality and intensity of the signal, as is usually done in studies reporting DAP- or CHIP-seq data.

Answer: Thank you for the suggestion. We added the actual peaks in Figure 3, 4.

- The authors have resequenced their bacterium to provide a more complete

assembly of the genome. As this new assembly is used as a reason for the need of reanalyzing the TCS regulon, the differences brought by this new assembly should be briefly discussed.

Answer: Thank you for the constructive suggestion, we added the information 'The previous version of genome assembly was consisted of 29 scaffolds (accession number: AENH00000000), so we subsequently re-sequenced RA-YM via PacBio platform to generate the completion map (accession number: NZ_CP079205). The long read-length of third generation sequencing and the small impact of GC content enabled us to solve the problem of repetitive sequences in RA-YM and avoid the unevenness of sequencing caused by GC content, which are useful for de novo genome assembly. The new version genome is 2,153,508 bp in length, with an average G+C content of 35.02%.' in Line 157-165(125-133).

Minor comments :

- Concerning the virulence phenotype, can the authors comment on what is the reason for the decreased virulence in the PhoPR mutants ? i.e. which target genes could be responsible ?

Answer: In the previous study, we found that RA-YM after the deletion of *phoPR* lost its pathogenicity, so we began to analyze the genes directly regulated by *phoPR* to try to screen and analyze whether there were any virulent genes directly

regulated by *phoPR*. Therefore, we performed the individual deletions of *phoP* and *phoR*, and performed RNA-seq and DAP-seq for PhoP. From analysis of genes directly regulated by *phoP*, we did not find traditional virulence genes, but we found some genes involved in the survival in the host, like Bat operon associated with aerotolerance. Another candidate target gene, DedA, has also attracted our attention, which is involved in virulence in the research of *K. pneumoniae* (<https://doi.org/10.1038/s41598-021-03834-3>) and *B. glumae* (<https://doi.org/10.1128/AEM.00915-21>). We will focus on Bat operon and DedA in the future. Of course, we will also expand the screening range of virulence genes to genes indirectly regulated by *phoP* or *phoR* (S3 table) later.

- Statistical analysis should be performed in figure 8

Answer: Thank you for the suggestion. We added the statistical analysis, and we added the P-value style in the figure legend, as 'p (***) < 0.0002, p (****) < 0.0001' .

- DAP-seq is an emerging method for the study of TF binding sites that is not yet well known, especially in studies on prokaryotes, attention should thus be paid to properly cite the method and previous works. Particularly, the authors do not cite papers from the O'Malley group, which developed DAP-seq, but only older versions of the DAP-chip approach. At least the Nature protocol DAP-seq paper should be cited (<https://doi.org/10.1038/nprot.2017.055>). Additionally, the authors use a version of the protocol that is made for response regulators and uses acetyl phosphate, which has already been used in other bacterial studies (ex: <https://doi.org/10.1093/nar/gkab928> ; <https://doi.org/10.1111/mmi.13909>) which should also be considered as references.

Answer: Thank you for the suggestion, we added the references.

- Line 462: "The present study has reported the first application of the combination of DAP-seq with RNA-seq ...": that claim is not true (see <https://doi.org/10.1111/mmi.13909> for an example of bacterial TCSs, and <https://doi.org/10.1128/mSystems.00753-20> for bacterial OCSs). Citing previous similar studies should thus be considered.

Answer: Thank you for the suggestion, we added the references and modified the sentence as 'The present study has reported the application of the combination of DAP-seq with RNA-seq to perform genome-wide identification of the two-component systems regulon in pathogenic bacteria.' in Line 573-575(459-461).

- Figure 5 could be made a supplementary figure.

Answer: Thank you for the suggestion. And we have modified according to the comment.

- Line 61 : Should remove dot between end of sentence and reference

Answer: Thank you for the suggestion, and we have removed the dot throughout the text.

- Should homogenize way of referring to papers in the text, ex: "Dou reported..." or "Tian et al. identified...". I suggest using et al. in all instances

Answer: Thank you for the suggestion, and we have homogenized way of

referring to papers throughout the text.

- Line 400 & 402: "phoP and phoP"

Answer: Thank you for the suggestions. According to the comment, we modified the sentences as 'Further, the single-gene deletion strains namely *ΔphoP* and *ΔphoR* respectively were successfully constructed. The growth curve showed that *ΔphoP* and *ΔphoR* have decreased growth rates at every time point compared to the WT' in Line 470-472(365-367).

- Generally, the manuscript should be thoroughly proofread again for typos and confusing phrasings

Answer: Thank you for the suggestion. We have proofread the manuscript carefully.

We appreciate for Editors/Reviewers' warm work earnestly, and hope that the correction will meet with approval. We look forward to hearing from you regarding our submission. We would be glad to respond to any further questions and comments that you may have. We appreciate for Editors/Reviewers' warm work earnestly, and hope that the correction will meet with approval. Once again, thank you very much for your comments and suggestions.

Prof. Zili Li

College of Veterinary Medicine

Huazhong Agricultural University

No.1, Shizishan Street

Wuhan, Hubei 430070

China

September 15, 2022

Prof. Zili Li
Huazhong Agricultural University
College of Veterinary Medicine
No.1, Shizishan Street
Wuhan, Hubei 430070
China

Re: Spectrum01883-22R1 (Genome-wide analysis reveals PhoP regulates pathogenicity in *Riemerella anatipestifer*)

Dear Prof. Zili Li:

Your manuscript has been accepted, and I am forwarding it to the ASM Journals Department for publication. You will be notified when your proofs are ready to be viewed.

Sincerely,

Artem Rogovskyy
Editor, Microbiology Spectrum

Journals Department
Supplement Figure and Table: Accept